# Cerebellum encodes and influences the initiation, performance, and termination of discontinuous movements in mice

Michael A Gaffield[1], Britton A Sauerbrei[2], Jason M Christie[1]*†

[1]Max Planck Florida Institute for Neuroscience, Jupiter, United States; [2]Case Western Reserve University School of Medicine, Cleveland, United States

**Abstract** The cerebellum is hypothesized to represent timing information important for organizing salient motor events during periodically performed discontinuous movements. To provide functional evidence validating this idea, we measured and manipulated Purkinje cell (PC) activity in the lateral cerebellum of mice trained to volitionally perform periodic bouts of licking for regularly allocated water rewards. Overall, PC simple spiking modulated during task performance, mapping phasic tongue protrusions and retractions, as well as ramping prior to both lick-bout initiation and termination, two important motor events delimiting movement cycles. The ramping onset occurred earlier for the initiation of uncued exploratory licking that anticipated water availability relative to licking that was reactive to water allocation, suggesting that the cerebellum is engaged differently depending on the movement context. In a subpopulation of PCs, climbing-fiber-evoked responses also increased during lick-bout initiation, but not termination, highlighting differences in how cerebellar input pathways represent task-related information. Optogenetic perturbation of PC activity disrupted the behavior by degrading lick-bout rhythmicity in addition to initiating and terminating licking bouts confirming a causative role in movement organization. Together, these results substantiate that the cerebellum contributes to the initiation and timing of repeated motor actions.

*For correspondence:
jason.m.christie@cuanschutz.edu

Present address: †University of Colorado School of Medicine, Aurora, United States

## Editor's evaluation

This study conducts physiological recordings in awake mice to reveal how cerebellar Purkinje cells convey temporal information about the onset and offset of ongoing movements. There is a growing appreciation for the cerebellum in planned behavior, but how it contributes to a spontaneously initiated volitional behavior remained unclear. This work provides important insights into the roles of cerebellar Purkinje cells in regulating rhythmic movements under volitional control.

## Introduction

Voluntary movement often encompasses repeated starts and stops of the same deliberate motor action enabling animals to achieve their goals. Because these discontinuous movements require a temporal structure, the brain is thought to generate a timing representation of salient motor events, such as the transition to movement initiation and/or termination, that improves the consistency of behaviors across repetitions (*Ivry et al., 2002*). This activity may assist in the preparation for voluntary movement, as goal-directed actions must be planned prior to initiation (*Ghez et al., 1991*). Interconnected brain regions, including the motor cortex, thalamus, basal ganglia, and cerebellum, play a role in planning and executing deliberate movements (*Gao et al., 2018*; *Guo et al., 2015*; *Kunimatsu et al., 2018*). Human studies have helped to elucidate the putative role of each brain structure in motor control. For example, patients with cerebellar damage often have difficulty in accurately timing

the initiation and termination of periodically performed discontinuous movements but can otherwise execute continuous rhythmic patterns of motor output in a relatively unimpaired manner (*Bo et al., 2008*; *Schlerf et al., 2007*; *Spencer et al., 2003*). These findings lend support to the idea that the cerebellum processes predictive information related to impending transitions to motor action and inaction such that planned movements are finely timed and thus well executed (*Bareš et al., 2019*; *Ivry et al., 2002*; *Tanaka et al., 2021*). Yet, experimental validation of this hypothesis at the neurophysiological level is lacking.

The cerebellar contribution to movement timing in the domain of sensorimotor prediction is frequently studied in animal models based on simple cue-evoked reflexive behaviors. For delay eyeblink conditioning, cerebellar activity begins ramping in response to sensory cues that predict an impending conditioned response (*Giovannucci et al., 2017*). Similar ramping of cerebellar activity occurs during other learned behaviors in which sensory cues provide a trigger for movement initiation (*Bina et al., 2020*; *Tsutsumi et al., 2020*; *Yamada et al., 2019*). Neural activity also ramps in the cerebellum during motor planning tasks in which a delay period precedes an impending deliberate movement (*Chabrol et al., 2019*; *Gao et al., 2018*; *Wagner et al., 2019*). As the delay time increases, the onset time of ramping activity shifts, with ramping activity commencing immediately prior to initiation, rather than throughout the entire delay (*Kunimatsu et al., 2018*; *Ohmae et al., 2017*). This ramping activity is influential in shaping behavior because its disruption can affect motor timing (*Ohmae et al., 2017*). Importantly, in many of these sensory-cue-driven tasks, animals must actively avoid executing the motor plan during the delay period because false starts are punished. Overall, the representation of both sensory cues and negative valence signals in cerebellar activity presents challenges in directly assessing how this brain region encodes and directly influences the timing of salient motor-event transitions.

To isolate motor-event-related neural activity in the mouse cerebellum, we trained mice to perform a periodic, discontinuous movement task requiring them to conduct a stereotyped behavior at a regular interval to acquire a reward. Movement-timing activity was apparent in Purkinje cells (PCs) located in the Crus I and II lobules of the lateral cerebellar cortices. At the population level, PC simple spike firing was related to movement kinematics and ramped immediately before the initiation of each cycle of motor action. The ramping onset times were earlier when the movement was internally triggered compared with cases in which the same action was elicited by a sensory cue. PC simple spiking also ramped immediately before the termination of each action cycle. By contrast, climbing-fiber-driven PC activity increased only prior to movement initiation. Optogenetic perturbation of PC activity disrupted movement rhythmicity, terminated ongoing movement, and could initiate movement when the optogenetic stimulus ended. Thus, the cerebellum plays an active role in the temporal organization of periodically performed discontinuous movements under volitional control.

## Results

### Mice learn to perform a discontinuous movement task based on internal timing

To understand how cerebellar activity relates to the organization of well-timed transitions to motor action and inaction during periodically performed movements, we trained head-fixed mice to self-initiate bouts of licking at regular intervals, where each bout consisted of rhythmic protractions and retractions of the tongue at 6–8 Hz (*Horowitz et al., 1977*). For this purpose, we used an interval timing task in which thirsty mice consumed water droplets dispensed at a fixed time interval (*Toda et al., 2017*). In the task structure (*Figure 1A*), we randomly withheld water allocation in 20% of the trials. Because the mice voluntarily initiated licking bouts without any sensory cues during these unrewarded trials, this step allowed us to assess their ability to elicit an internally planned, periodic motor behavior that anticipated the regular timing of water-reward availability.

In the first few sessions of task performance (fixed time interval of 10 s), beginner mice typically licked sporadically throughout each trial without regard to the timing of water allocation (*Figure 1B, C*). With experience, the mice learned to alter their strategy to concentrate licking bouts in response to water delivery (*Figure 1D*). Trained mice eventually initiated exploratory licking bouts prior to water-reward delivery and terminated licking shortly after consuming the dispensed water droplet resulting in, generally, just one bout per trial (*Figure 1E, F*). This behavioral change occurred without

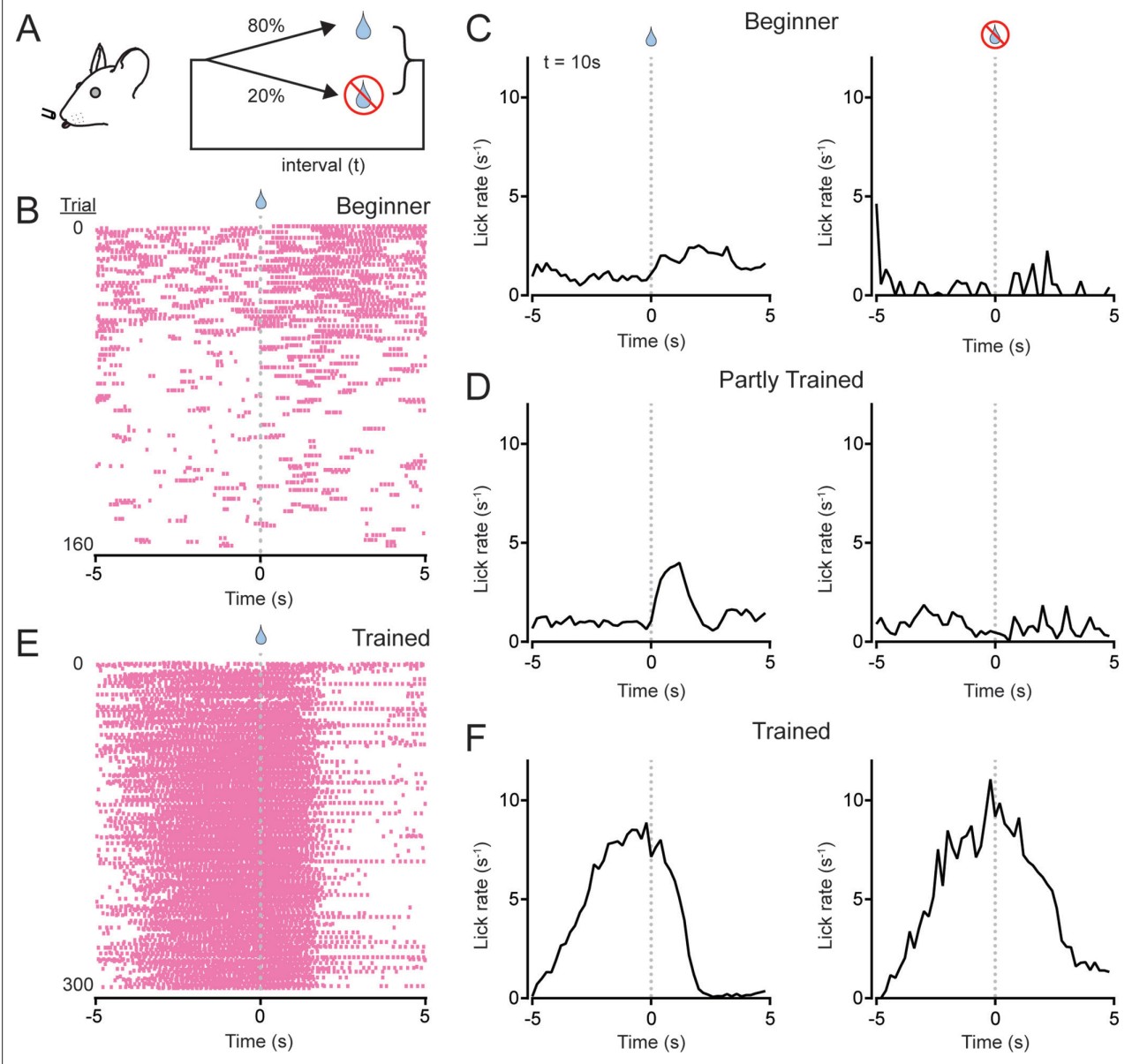

**Figure 1.** An interval timing task to assess the role of the cerebellum in organizing periodic, discontinuous movement. (**A**) Schematic diagram of the task. Mice were trained to lick for water rewards delivered at a regular time interval (*t*). Water was allocated in most trials and withheld in the others. (**B**) Lick patterns of a beginner mouse during an early training session. Licks are indicated by pink tic marks; water was allocated at the time indicated by the droplet (*t* = 10 s). (**C**) Trial-averaged lick rates for the beginner mouse with session trials (159 total) separated based on whether water was allocated (left; rewarded) or withheld (right; unrewarded). (**D**) Same as panel C but after the mouse received additional sessions of training (216 total trials). (**E**) Lick patterns over the course of a session after full training (same mouse as in panels B and C). (**F**) Trial-averaged lick rates of the fully trained animal during an individual session (300 total trials).

an overt punishment to actively suppress licking during the delay period. In trained mice, the licking behavior was essentially unchanged during water-omission trials, demonstrating their ability to anticipate reward timing and withhold their behavior until the next trial (*Figure 1F*). These results show that the mice reliably self-initiate regular bouts of voluntary licking based solely on internal timing, presumably referenced by the amount of elapsed time since the previous water reward, and abruptly stop licking after consuming the water reward for each trial (*Rossi et al., 2016*; *Toda et al., 2017*).

## Engagement of the cerebellum during internally timed discontinuous movements

To identify neural correlates of task-related behavioral events in the cerebellar cortex, we used extracellular electrophysiology to record from cells in the lobules of the left Crus I and II (*Figure 2A*), regions of the lateral cerebellum implicated in orofacial behaviors (*Bryant et al., 2010*; *Welsh et al., 1995*). Although neuronal population activity was densely sampled using multielectrode silicon probes, we restricted our analysis to PCs because they form the sole channel of output from the cerebellar cortex. We identified putative PCs based on their location, firing characteristics, and size (see Materials and methods; *Figure 2—figure supplement 1*; *Tsutsumi et al., 2020*).

In trained mice, PCs in both lobules displayed a heterogeneous range of activity changes in simple spiking patterns during task performance (*Figure 2A*). In water-rewarded trials, the average activity of Crus I PCs increased as mice began exploratory licking in anticipation of water delivery. The mean simple spike rate increased further once the mice detected water and began consummatory licking (*Figure 2B*). In unrewarded trials, the PC population activity showed a similar increase during exploratory licking (*Figure 2C*). However, in these trials, PC simple spiking lacked a prominent second peak in activity. The average activity pattern of Crus II PCs closely resembled that of Crus I PCs. The simple spiking rate increased during exploratory licking and evolved with a further sharp uptick as consummatory licking commenced (*Figure 2D, E*).

The uptick in PC simple spiking during consummatory licking, relative to exploratory licking, may be attributable to the encoding of reward acquisition, which has been shown to be represented in the activity of granule cells (*Wagner et al., 2017*). Yet, the overall simple spiking rate showed a linear correspondence to the licking rate when assessed across all contexts of the task in either Crus I or II PCs (*Figure 2—figure supplement 2*). Therefore, the elevation of PC simple spike firing in response to reward allocation may instead reflect the abrupt increase in licking rate during water consumption (*Figure 2B, D*). To further explore the possibility that PC activity was related to the context of reward, we separated licking trials dependent on whether water was allocated at the expected interval or at an unexpectedly prolonged interval resulting from reward omission on the immediately preceding trial (*Figure 2—figure supplement 3*). The pattern of PC activity in the Crus I PC ensemble appeared essentially the same in these two trial types (*Figure 2—figure supplement 3*) indicating that the PC spike representation of putative motor-related information was not disrupted because of the prior reward-expectation error. In addition, we did not observe persistent alterations in PC activity on water allocation trials during which the mice did not choose to lick (*Figure 2—figure supplement 4*). Based on these results, we conclude that PCs in both Crus I and II are similarly engaged during the performance of internally timed bouts of voluntary licking and that their ensemble activity largely form a representation of movement.

Given that PCs appeared to encode motor parameters, we next evaluated whether some of the PC activity reflected encoding of individual licks that comprise licking bouts by aligning simple spikes of each cell to cycles of tongue protrusion and retraction (see Materials and methods). Many PCs showed a clear phasic modulation of their average firing during the lick cycle (*Figure 3A*). The timing of the peak in spiking activity, relative to the lick cycle, varied among PCs (*Figure 3B, C*). Thus, there is a near continuum of lick-phase representation in the collective responses of Crus I and II PCs. Although nearly all PCs were significantly entrained to the lick cycle (92%), the depth of lick-phase-modulated simple spike activity varied widely across the population (*Figure 3D*). Interestingly, there was no obvious relationship between the entrainment strength of PC firing to the lick cycle and the average change in firing rate of PCs around the time of water allocation when the licking rate was greatest (*Figure 3E*). In conclusion, Crus I and II PCs form a fast-timescale representation of the licking rhythm that is nested in a broader representation of additional motor variables related to the overall movement. To ascertain whether some of the broader PC activity in these regions specifically encodes transitions that delimit discontinuous periodic movements, we refined our analysis to examine simple spike firing around two salient motor events: lick-bout initiation and lick-bout termination.

## PC simple spiking modulates during the transition to movement initiation

For voluntary movements, motor plans are converted into motor actions at the time point of initiation. In our task, this transition occurred at lick-bout onset, when mice began rhythmic licking. To examine

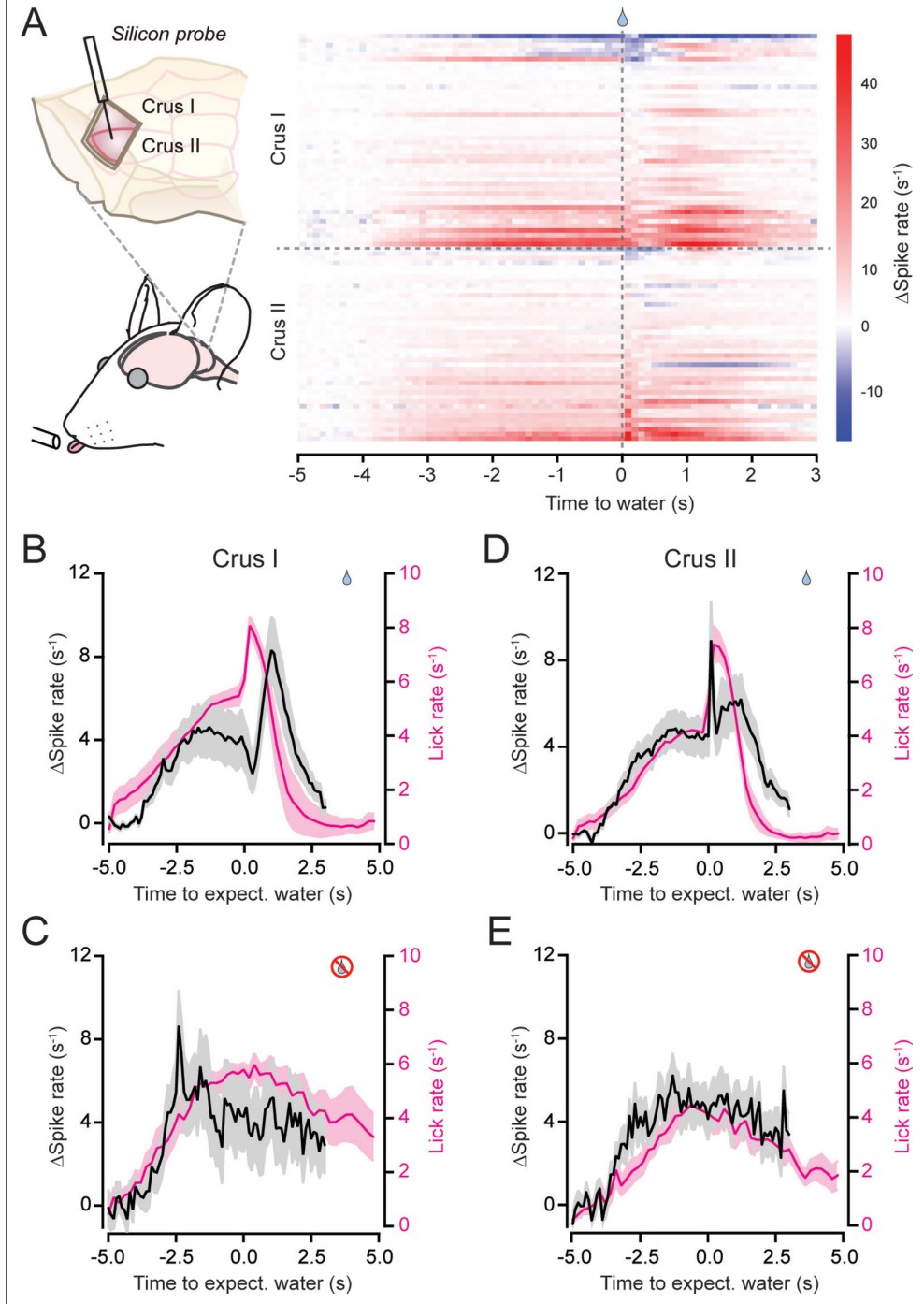

**Figure 2.** Modulation of Purkinje cell (PC) simple spiking during the performance of discontinuous movement.
(**A**) Left: electrophysiological activity was recorded from PCs using silicon probes targeting either the left Crus
I or II through a large craniotomy. Right: changes in simple spiking firing, relative to non-licking baseline, for
all PCs during water-rewarded trials. Data are separated by lobule and sorted based on average activity-level
changes within ±200 ms of water allocation (n = 47 Crus I PCs from 6 mice; n = 42 Crus II PCs from 5 mice). (**B**)
The mean change in simple spike rate (black), relative to baseline, for Crus I PCs during water-rewarded trials. The
corresponding trial-averaged lick rate is also shown (pink). (**C**) Same as panel B but for trials in which water was
withheld. (**D, E**) Same as panels B and C but for Crus II PCs.

The online version of this article includes the following figure supplement(s) for figure 2:

**Figure supplement 1.** Identification of Purkinje cells (PCs) in silicon probe data.

*Figure 2 continued on next page*

*Figure 2 continued*

**Figure supplement 2.** Movement-related Purkinje cell (PC) simple spiking activity.

**Figure supplement 3.** Purkinje cell (PC) simple spiking activity does not change in response to unexpected intervals.

**Figure supplement 4.** Lack of nonmotor Purkinje cell (PC) simple spiking activity in nonmovement trials.

the simple spiking pattern at the time point of movement initiation, we aligned PC activity to the first lick of well-separated bouts (i.e., bouts with ≥2 s of preceding nonlicking) in individual trials across animals (*Figure 4A*), thus sharpening our ability to discern the temporal correspondence of PC firing and the start of the behavior. Following this event-triggered averaging, we observed that the simple spiking rate generally began to increase several hundred milliseconds before the detection of the

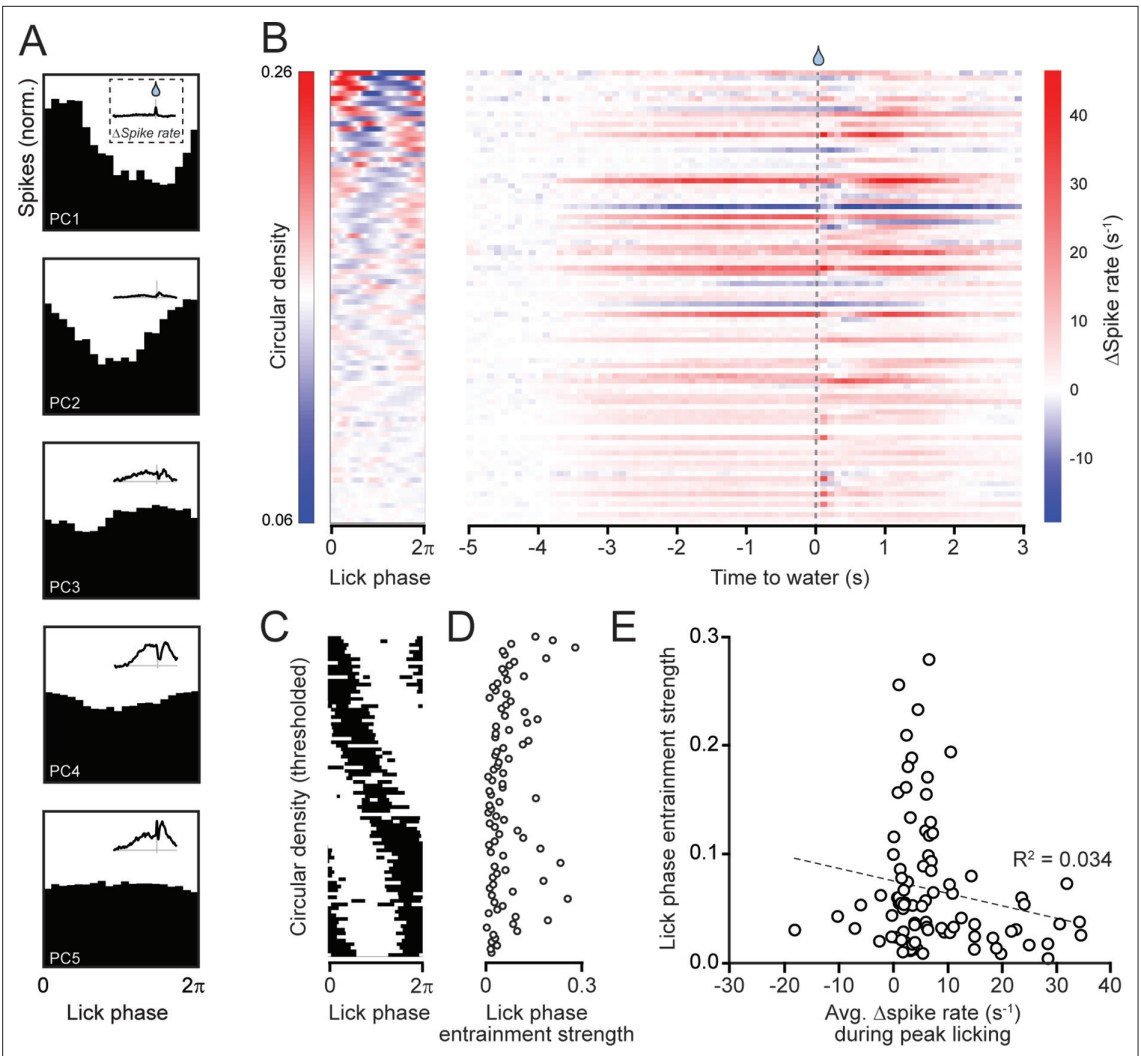

**Figure 3.** Entrainment of Purkinje cell (PC) simple spiking to the licking rhythm. (**A**) In spike histograms from five example PCs, lick phases sampled at spike times show varying levels of modulation. Contact between the tongue and the water port is defined to be 0 = 2π. The upper insets show the smoothed firing rate profiles of the same PC over the entire trial epoch, aligned to the time point of expected water allocation. (**B**) Probability density for spikes over the lick cycle (left) and the change in firing rate over the entire trial (right; reordered from *Figure 2B*). Each row corresponds to a single PC and is sorted by the strength of entrainment to licking, defined as the mean resultant length. (**C**) Thresholded probability density for spikes over the lick cycle. Black regions indicate phases at which the density exceeds 1.02/2π. Rows are sorted by the entrainment phase, defined as the angle of the mean resultant. (**D**) Entrainment strength for all PCs, sorted as in C. (**E**) The relationship between firing rate modulation following water delivery and the strength of entrainment to the licking rhythm for individual PCs. The change in simple spike firing rate was averaged over the 2 s following water delivery. Dashed line is the linear correlation. Note, one outlier PC is off scale in panels D and E.

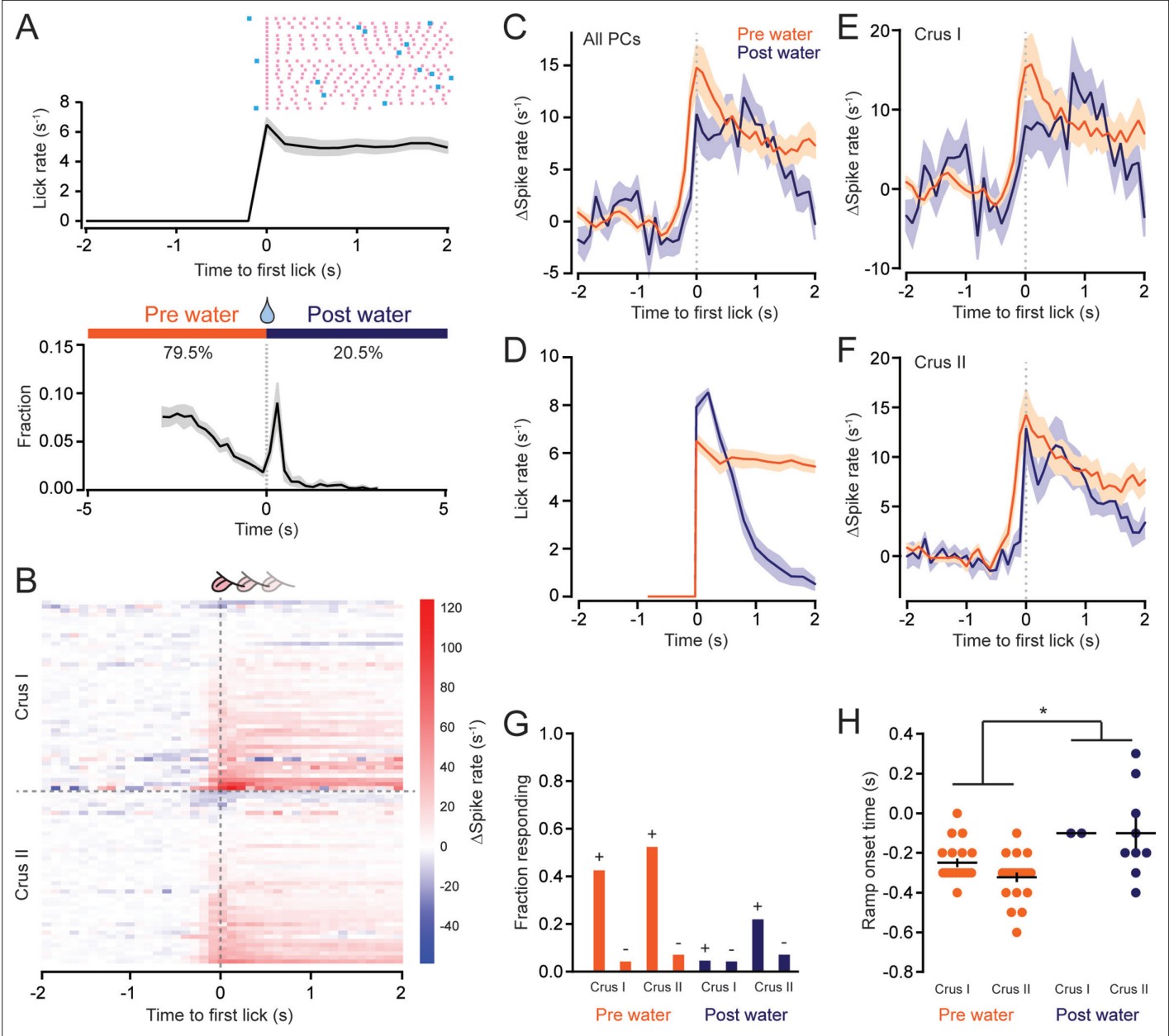

**Figure 4.** Modulation of Purkinje cell (PC) simple spiking during the initiation of discontinuous movement. (**A**) Top: plot of the mean lick rate aligned to the time point of the first lick in each water-rewarded trial bout (n = 3164 trials of well-isolated licking bouts from 11 mice). The lick patterns for example trials of an individual mouse are also shown with licks indicated by pink tic marks and the time of water allocation indicated in blue. Bottom: histogram of lick-bout initiation times relative to water allocation. For most trials, mice began exploratory licking prior to water delivery (pre water). However, in the remaining trials, mice refrained from licking until after water allocation (post water), which immediately triggered a rapid increase in lick-bout initiations to consume the dispensed droplet. (**B**) The change in simple spiking activity relative to baseline for individual PCs sorted based on their average activity levels within ±200 ms of the first lick in each water-rewarded trial bout. (**C**) Trial-averaged change in simple spike activity for PCs, separated depending on whether mice initiated lick bouts before or after water allocation (pre and post water, respectively). Note the ramps in activity prior to licking. (**D**) Licking rates for bouts separated whether licking began before or after water allocation. (**E, F**) Same as panel C expect for Crus I (panel E) or Crus II (panel F) PCs. (**G**) Fraction of individual Crus I and II PCs with activity profiles that were either positively (+) or negatively (−) modulated around the time of lick-bout initiation, separated depending on whether licking began before or after water allocation (pre and post water, respectively). (**H**) Comparison of the onset times of activity ramping, relative to the first lick in trial bouts, for PCs whose activity positively modulated around lick-bout initiation. Data from each lobule were grouped together for statistical comparison (pre water: n = 42 PCs from 10 mice; post water: n = 11 PCs from 7 mice). Asterisk indicates significance (p = 0.0129, Student's t-test). See also *Figure 4—source data 1*.

The online version of this article includes the following source data and figure supplement(s) for figure 4:

*Figure 4 continued on next page*

*Figure 4 continued*

**Source data 1.** Source data for *Figure 1H*.

**Figure supplement 1.** Purkinje cell (PC) simple spiking firing at lick-bout initiation.

**Figure supplement 2.** Simple spike firing rates of Purkinje cells (PCs) at lick-bout initiation.

**Figure supplement 3.** Simple spiking profiles of representative Purkinje cells (PCs) to lick-bout initiation.

first lick in a bout (*Figure 4B*) with the onset time of the average PC population response leading the first lick by 240 ± 40 ms (*Figure 4—figure supplement 1*). Because lick initiation is rapid – the time between any visible mouth movement to full tongue protrusion is 30–60 ms (*Bollu et al., 2021*; *Gaffield and Christie, 2017*) – this ramping is more suitable for reflecting preparatory activity rather than an online representation of movement execution.

As noted above, the precise timing of lick-bout initiations, relative to water availability, varied from trial to trial (*Figure 4A*). As expected for trained mice that accurately anticipate impending rewards, most licking bouts were exploratory, beginning prior to water allocation (*Figure 4A*). However, in some trials, the mice did not perform any exploratory licking and, instead, waited until water became available before immediately commencing a bout of consummatory licking (*Figure 4A*). It is unclear why the mice withheld their licking until after water delivery during these trials. However, the tight distribution of lick-bout initiations around the time of water allocation in these trials (median response time: 360 ± 110 ms) indicates that the ensuing consummatory movements were reactive and were likely triggered by sensory evidence indicating water availability (i.e., the mice performed a licking bout after they detected the presence of water).

Separating trials of PC activity based on whether licking bouts were initiated before or after water allocation led to an unexpected result. The average simple spiking rate of PCs began to ramp earlier for exploratory licking bouts, when the movements were initiated prior to water allocation, compared to that of bouts in which consummatory licking commenced immediately after water became available (*Figure 4C*). Although mice exhibited elevated licking rates for reactive licking bouts relative to exploratory licking bouts (*Figure 4D*; peak rates: 6.5 ± 0.3 and 8.5 ± 0.2, pre- and post-water licking bouts, respectively; p < 0.001, Student's *t*-test), there was no difference between the peak rates of PC simple spike firing for the two licking contexts (*Figure 4C*; peak Δspike rate within 500 ms of lick onset: 14.8 ± 2.1 and 10.3 ± 2.1, pre- and post-water licking bouts, respectively; p = 0.13, Student's *t*-test). We observed a consistent temporal advance in simple spike ramping activity for exploratory licking bouts compared to water-reactive licking bouts for both Crus I and II PCs (*Figure 4E, F*, *Figure 4—figure supplement 2*). There were no differences in the peak change in simple spike firing rate for either Crus I PCs (peak Δspike rate: 15.7 ± 3.8 and 8.4 ± 3.3, pre- and post-water licking bouts, respectively; p = 0.15; Student's *t*-test) or Crus II PCs (peak Δspike rate: 14.3 ± 2.6 and 12.9 ± 2.8 pre- and post-water licking bouts, respectively; p = 0.72; Student's *t*-test) around the time of peak licking (*Figure 4E, F*).

These results indicate a robust activation of the PC ensemble in the lateral cerebellum prior to lick-bout initiation. However, there was heterogeneity in the simple spike response patterns of individual PCs. Some PCs positively modulated their firing during lick-bout initiations relative to their baseline firing rate (*Figure 4—figure supplement 3*), whereas other PCs negatively modulated their firing (*Figure 4—figure supplement 3*); the few remaining PCs were unresponsive during this motor transition. As expected from the population average, PCs that positively modulated their simple spike activity were more common than PCs that negatively modulated their activity (*Figure 4G*). Additionally, PCs were more likely to positively modulate their activity prior to exploratory licking bouts (i.e., ramp) that preceded water allocation than for bouts of consummatory licking that were reactive to water allocation (*Figure 4G*).

In the subset of PCs with positively modulated activity, a comparison of ramping onset times across conditions revealed that activity began ~200 ms earlier for bouts of exploratory licking compared with bouts of purely reactive consummatory licking (*Figure 4H*). However, among the PCs that positively responded to both licking contexts (*Figure 4—figure supplement 3*), the onset time of activity ramping for anticipatory versus reactive licking was not different (−300 ± 114 and −113 ± 242 ms, respectively; *n* = 8; p = 0.07; Student's *t*-test). This implies that the timing difference in ramping activity in positively modulating PCs must also involve the participation of cells with different types of tuning (e.g., to either exploratory or reactive licking alone; *Figure 4—figure supplement 3*). We

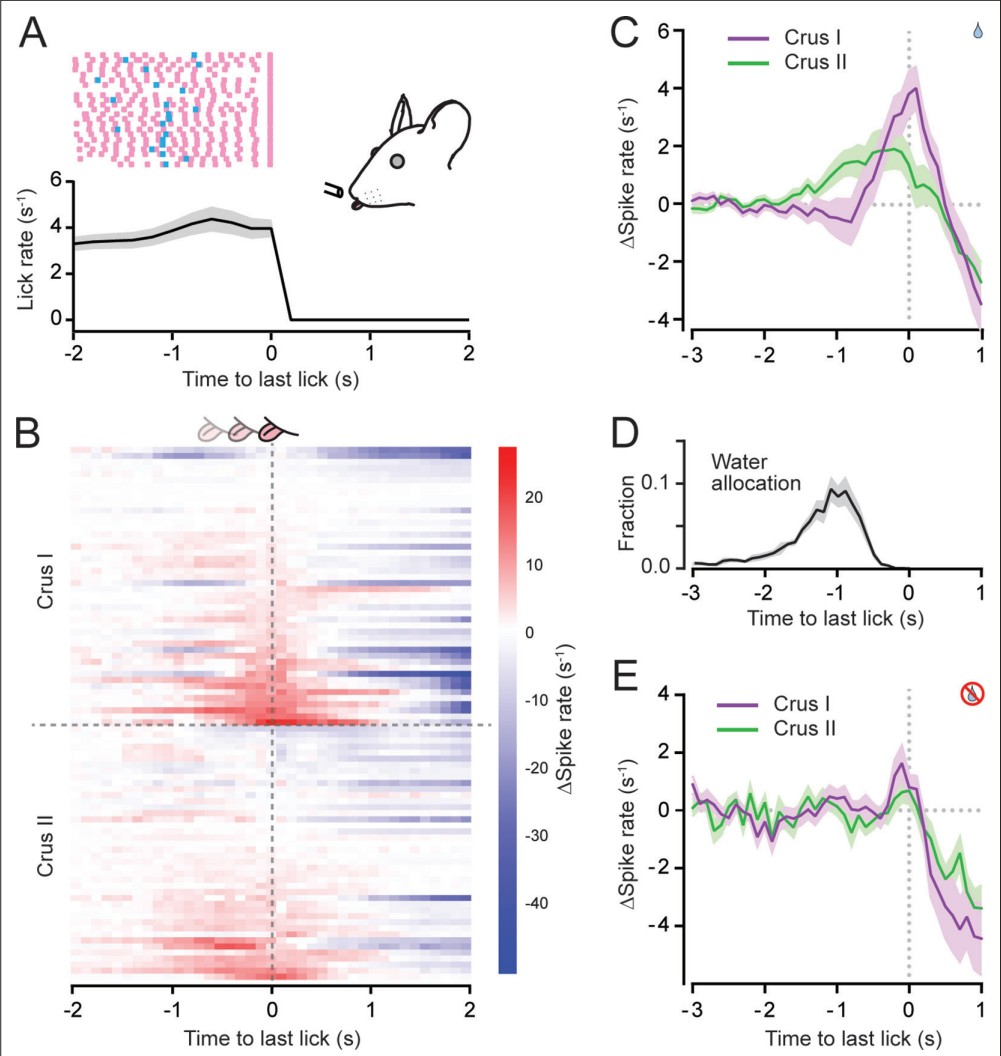

**Figure 5.** Modulation of Purkinje cell (PC) simple spiking during the termination of discontinuous movement. (**A**) Plot of mean lick rate aligned to the last lick in water-rewarded trial bouts ($n$ = 2846 trials from 11 mice). Also shown are lick patterns of example trials for an individual mouse with licks indicated by pink tick marks and the timing of water allocation in blue. (**B**) Change in simple spike activity for individual PCs sorted based on their average activity levels within ±200 ms of the last lick in each water-rewarded trial bout. Responses were baselined to the lick-related activity in the seconds prior to the last lick. (**C**) Trial-averaged change in simple spike activity for Crus I ($n$ = 47 from 6 mice) and Crus II ($n$ = 42 from 5 mice) PCs aligned to the time of the last lick in water-rewarded trials. (**D**) Distribution of the timing of water allocation aligned to the point of the last lick in rewarded trial bouts (same trials as in panel A). (**E**) Same as panel C but for unrewarded trials ($n$ = 44 Crus I PCs from 6 mice; $n$ = 38 Crus II PCs from 5 mice).

The online version of this article includes the following figure supplement(s) for figure 5:

**Figure supplement 1.** Last-lick-related Purkinje cell (PC) activity.

**Figure supplement 2.** Task-feature tuning of individual Purkinje cells (PCs).

conclude that PC simple spike rates change prior to the initiation of licking bouts, with the timing of ramping activity depending on whether the ensuing movement was initiated by internal motivation or was triggered by a sensory cue indicating water availability.

## PC simple spiking modulates during the transition to movement termination

The completion of a motor action is also a salient event. Therefore, to investigate whether PCs encode information pertaining to movement termination, we aligned PC simple spike activity to the last lick of a lick bout (*Figure 5A, B*). PCs displayed heterogeneous output patterns during this motor transition, with some cells positively modulating their activity and others negatively modulating their activity (*Figure 5B*). On average, the simple spiking rate in the PC population abruptly increased just prior to the last lick for cells in both Crus I and II during water-rewarded trials (*Figure 5C*; *Figure 5—figure supplement 1*). Because the last lick in a bout occurred well after the time point of water delivery (median time interval: 1.37 ± 0.21 s; *Figure 5D*), most of this ramping activity occurred after the dispensed water droplet had been consumed. In addition to encoding the impending termination of a lick bout, this simple spiking increase may also represent information pertaining to swallowing and/ or reward signaling by satiation in the gastrointestinal system (*Augustine et al., 2019*; *Zimmerman et al., 2019*). However, simple spiking also ramped prior to the last lick in bouts elicited during water-omission trials (*Figure 5E*; *Figure 5—figure supplement 1*), albeit at a much-reduced level compared to the end of water consumption trials. Thus, the ramping activity in PCs at the end of licking bouts most likely corresponds to the anticipation of motor-action termination. Because the level of licking diminished from a greater maximal rate during consummatory licking relative to anticipatory licking in the unrewarded condition (see *Figure 2B–E*), the differences in ramping PC activity at lick-bout termination between the two licking contexts could reflect differences in the level of overall movement prior to the cessation of action which is reflected in low levels of task-related PC activity at that time point (*Figure 5—figure supplement 1*). Overall, lick-bout termination was well represented in the PC activity in both lobules (36.2% and 23.8% of PCs in Crus I and II, respectively, modulated their simple spiking [see Materials and methods]).

To examine whether individual PCs tune their activity to specific task features, we sorted all cells based on how their simple spiking modulated during motor-event transitions. PCs were categorized depending on whether they exhibited ramp firing to exploratory licking initiated prior to water allocation, ramp firing to consummatory licking initiated after water allocation, and/or ramp firing at the end of either type of licking bout. Because negatively modulated responses were relatively rare, we pooled PCs that exhibited decreased firing during any motor transition into a single group. Plots of mean spiking rates from several of these groupings, including PCs that positively modulated their firing around either the first or last lick in a bout or that negatively modulated their firing (*Figure 5— figure supplement 2*), confirmed that our sorting differentiated PCs based on their response profiles. Overall, individual PCs in both Crus I and II were heterogeneous in their representation of task-related attributes (*Figure 5—figure supplement 2*). Although specialist PCs were common, for example, showing a preference for ramp firing at the initiation of exploratory licking, many PCs were engaged by multiple types of motor transitions, for example by increasing their firing to both licking initiation and termination. In summary, the simple spiking activity of PCs in the lateral cerebellar cortex modulates in response to salient motor events during discontinuous bouts of periodic movements, with only modest tuning for a specific type of motor transition.

## Climbing-fiber-induced PC activity increases during movement initiation

In addition to simple spikes, PCs also fire complex spikes and simultaneous bursts of dendrite-wide calcium action potentials in response to excitation provided by climbing fibers, the axonal projections of inferior olive neurons (*Llinás and Sugimori, 1980*; *Ozden et al., 2009*). Therefore, to determine the representation of climbing-fiber-induced PC activity during periodically performed discontinuous movements, we used two-photon imaging to measure climbing-fiber-evoked dendritic calcium events in PCs expressing the calcium sensor GCaMP6f (*Figure 6A*). This imaging-based approach is more sensitive in detecting climbing-fiber-evoked activity because it is challenging to reliably distinguish all complex spike waveforms for individual PCs in extracellular electrophysiological unit recordings (*Sedaghat-Nejad et al., 2021*; *Tsutsumi et al., 2020*). In quiescent mice, individual calcium events

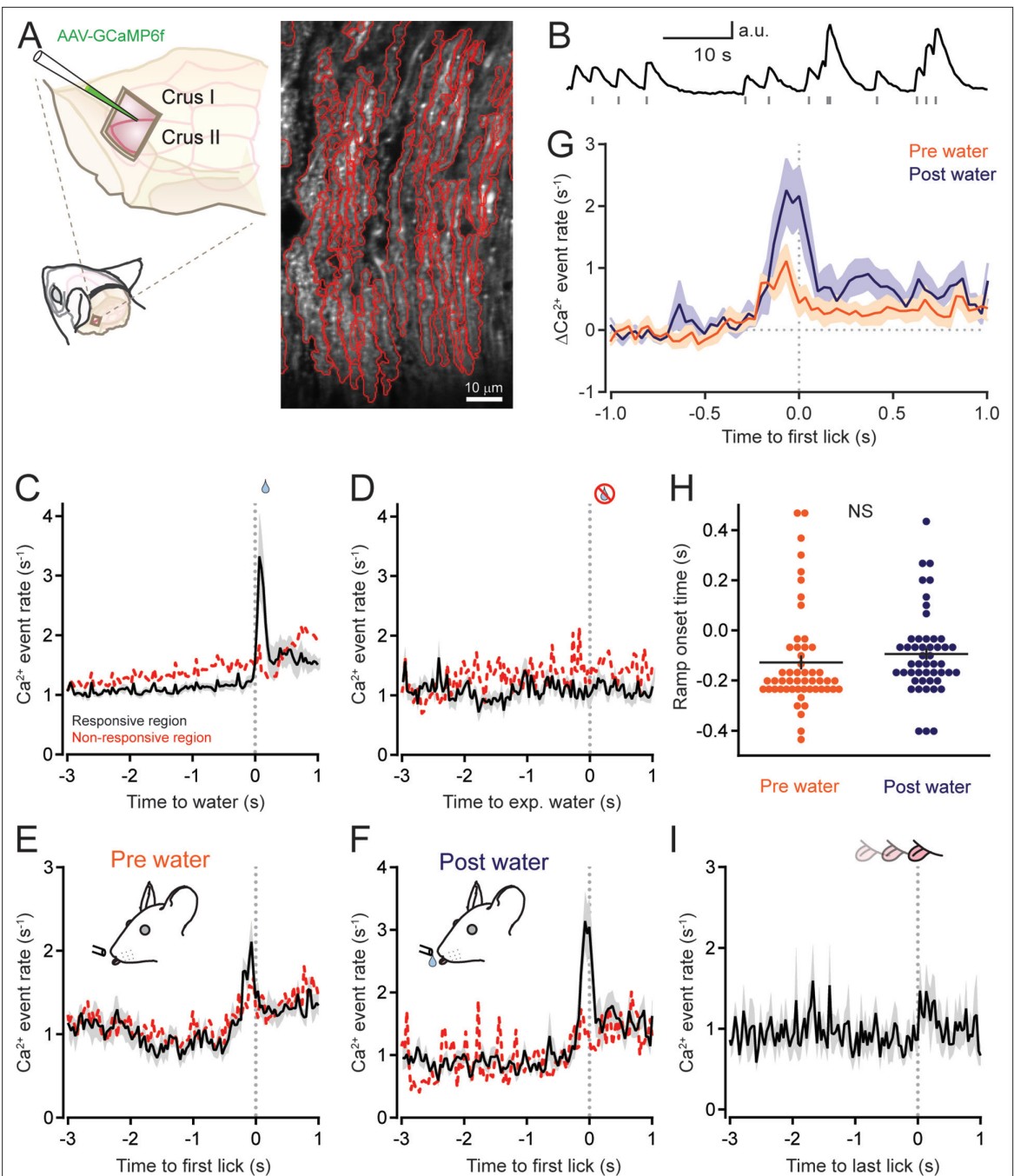

**Figure 6.** Climbing-fiber-evoked Purkinje cell (PC) activity increases at the initiation of discontinuous movement. (**A**) Left: AAV containing GCaMP6f under control of the *Pcp2* promoter was injected into Crus I and II to specifically transduce PCs; a cranial window provided optical access to the infected region. Right: a two-photon image with identified PC dendrites outlined in red. (**B**) Example fluorescence trace from a PC dendrite showing spontaneous calcium activity during quiescence. Individual climbing-fiber-evoked calcium events are indicated by gray tic marks. (**C**) Average climbing-fiber-evoked calcium event rates aligned to the time point of water delivery for water-rewarded trials. PC dendrites in some regions of Crus I and II showed clear increases in activity when mice elicited bouts of licking to water allocation (black line, *n* = 377 PCs in 8 ROIs from 4 mice), whereas in other regions there was very little to no change in activity (dashed red line, *n* = 239 PCs in 5 ROIs from 4 mice). (**D**) Same as panel C but for licking during unrewarded, water-omission trials. (**E**) Trial-averaged calcium event rates in PC dendrites aligned to the timing of the first lick in exploratory bouts initiated prior to water allocation (same data as panel C). (**F**) Same as panel E but aligned to the first lick for bouts of consummatory licking initiated after water allocation. (G) Top: overlay of the change in trial-averaged calcium event rates, relative to nonlicking baseline, for PCs in task-responsive regions of Crus I and II, aligned to the first lick of bouts initiated before or after water allocation (pre and post water, respectively). (**H**) Comparison of onset times for climbing-fiber-evoked calcium event ramping for individual PCs in trials where licking was initiated before (pre water; *n* = 52 PCs) or after (post water;

*Figure 6 continued on next page*

*Figure 6 continued*

*n* = 49 PCs) water allocation (see Materials and methods). Black line shows the mean (not significant, NS; p = 0.36, Student's *t*-test). (**I**) Trial-averaged calcium event rate aligned to the timing of the last lick in trial bouts (*n* = 616 PCs, *n* = 12 sessions, 5 mice). See also *Figure 6—source data 1*.

The online version of this article includes the following source data for figure 6:

**Source data 1.** Source data for *Figure 6H*.

were readily apparent in the dendrites of left Crus I and II PCs (*Figure 6B*), reflecting that climbing fibers continuously bombard PCs at 1–2 Hz (*Gaffield et al., 2016*; *Mukamel et al., 2009*; *Ozden et al., 2012*). However, during task performance, only a subset of the PCs showed a behavior-induced change in activity. This evoked response appeared to be spatially organized and dependent on the behavioral context. In water-rewarded trials, the average rate of climbing-fiber-evoked dendritic calcium events increased in PCs in some imaged regions of Crus I and II around the time of water allocation. However, in other imaged regions, there was no change in average PC calcium activity (*Figure 6C*). Interestingly, the increase in climbing-fiber-evoked activity during water consumption was absent in the same PCs during water-omission trials, when mice performed licking bouts but did not receive water rewards (*Figure 6D*). Thus, climbing fibers appear to signal reward-acquisition-related information to PCs in specific regions of Crus I and II (*Heffley and Hull, 2019*; *Heffley et al., 2018*; *Kostadinov et al., 2019*).

To evaluate the correspondence of climbing-fiber-evoked activity in PCs more carefully around motor-event transitions, we aligned dendritic calcium activity to the first licks in bouts of either exploratory licking initiated prior to water allocation or reactive licking initiated after water delivery. Average calcium event rates ramped prior to movement initiation for both licking contexts. These increases in climbing-fiber-evoked activity were prominent in the reward-responsive regions of Crus I and II (*Figure 6E, F*). Although the mice licked at a higher peak rate for reactive bouts than for exploratory bouts, the peak change in dendritic calcium event rates was not different between these licking contexts (*Figure 6G*; peak Δcalcium event rate: 1.12 ± 0.27 and 2.26 ± 0.60, pre- and post-water bouts, respectively; p = 0.113, Student's *t*-test). The onset times of calcium event ramping, relative to the detection of the first lick, were similar for the initiation of both exploratory and reactive bouts of licking (*Figure 6H*). Aligning PC dendritic calcium events to the last lick of lick bouts did not reveal any clear change in climbing-fiber-evoked activity around the transition to action completion (*Figure 6I*; calcium event rate 0.92 ± 0.09 and 1.28 ± 0.21 Hz prior to and immediately after the last lick; p = 0.2334, Wilcoxon signed rank test). Together, these results indicate that climbing-fiber-induced activity in a specific population of PCs ramps prior to the initiation, but not the termination, of both internally timed and sensory-cued bouts of goal-directed motor behavior.

## Optogenetic PC stimulation disrupts movement rhythmicity and both initiates and terminates action bouts

Having established that PCs in the lateral cerebellar cortex modulate their activity during the performance of discontinuous periodic movements, we applied an optogenetic approach to examine the causal role of PC activity in coordinating cycles of tongue protrusion and retraction, as well as motor-event transitions between action and inaction. We obtained conditional expression of channelrhodopsin-2 (ChR2) in all PCs by crossing transgenic Ai27 mice with the *Pcp2*^Cre driver line (*Madisen et al., 2012*; *Zhang et al., 2004*). In these animals, PC photostimulation drove robust simple spike firing, as measured in vivo using extracellular unit recording under anesthesia (*Figure 7A, B*). To optogenetically perturb PC activity during task performance, we bilaterally implanted optical fibers above the left and right Crus II lobule. Light stimuli were introduced during the period of peak anticipatory licking in a randomized subset of water-omission trials when trained mice were robustly engaged in performing internally timed, exploratory movements without any sensory evidence indicating water availability (*Figure 7C, D*). In response to PC photostimulation, the licking rate slowed and became erratic, showing a sharp degradation in rhythmicity (*Figure 7D, E*). Eventually, most mice ceased licking altogether (*Figure 7D*). After the PC photostimulation period ended, the mice resumed licking on some trials (51.4% ± 7.7%), albeit at a diminished rate compared with control trials (*Figure 7D*). Unilateral photostimulation of PCs in either the left Crus I or II lobule led to less dramatic effects on licking rhythmicity and rate (*Figure 7E*, *Figure 7—figure supplement 1*). The light stimulus had no

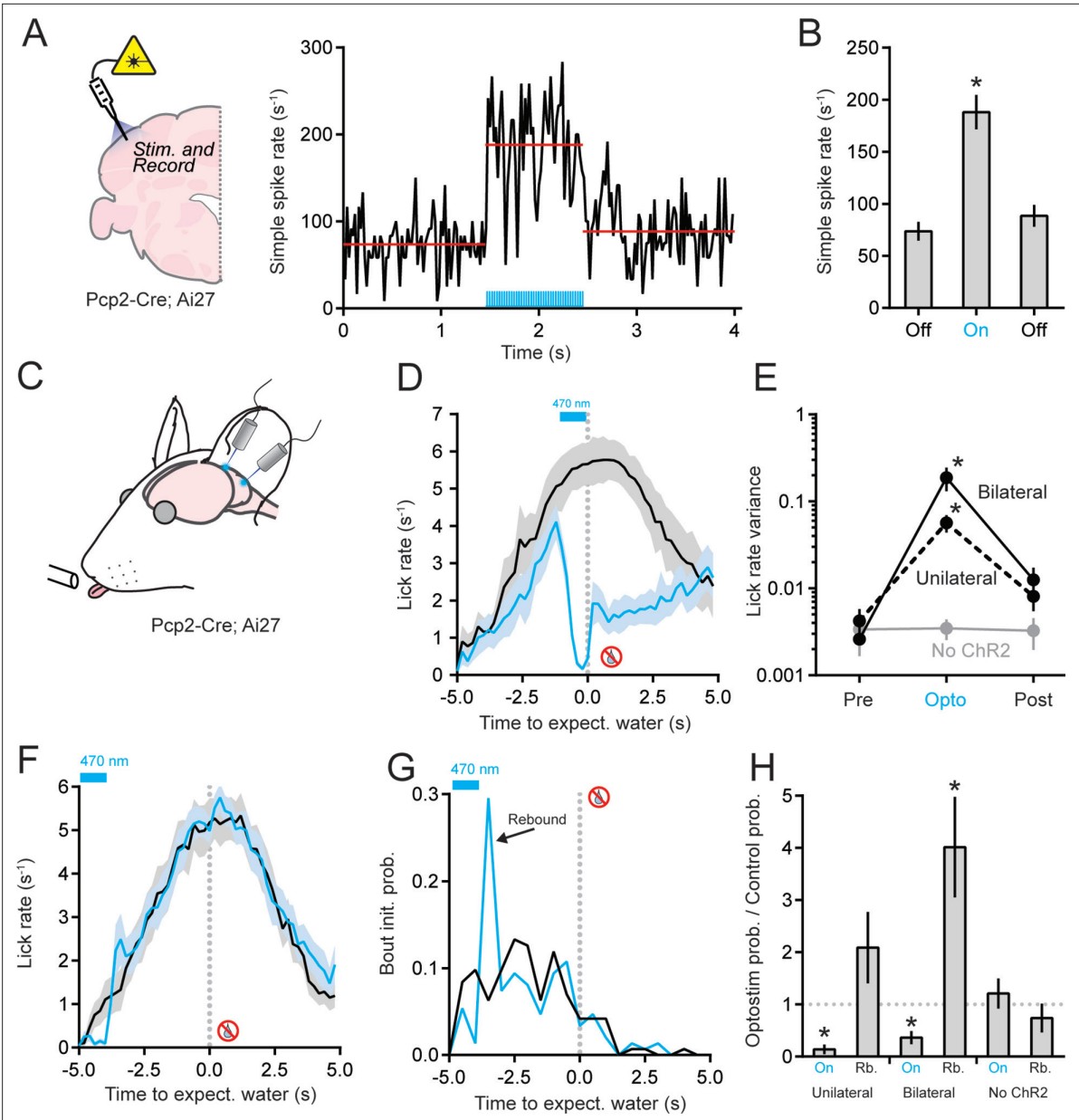

**Figure 7.** Optogenetic perturbation of Purkinje cell (PC) activity degrades the performance of discontinuous movements. (**A**) Left: extracellular electrophysiological measurements were obtained from ChR2-expressing PCs in response to photostimulation. Right: optogenetically induced simple spiking in a PC (light pulses indicated in blue). Red lines show the means for each epoch. (**B**) Summary plot of mean simple spike rate across PCs (*n* = 6) before, during, and after the optogenetic stimulus. Asterisk indicates significance ($p < 0.0001$, ANOVA with Tukey's post-test). (**C**) In mice with ChR2-expressing PCs, Crus II was bilaterally photostimulated during the interval task. (**D**) The effect of bilateral optogenetic PC activity perturbation on lick rate (blue) in unrewarded trials (*n* = 9 sessions, 3 mice). The photostimulus was timed to the period of peak licking, as referenced by interleaved control trials (black). Only well-isolated bouts were included in the analysis. (**E**) Summary of lick rate variability during optogenetic perturbation of PC activity. The *y*-axis is scaled logarithmically. Data include trials with bilateral photostimulation of Crus II (*n* = 9 sessions, 3 mice), trials with unilateral photostimulation of Crus I or II (*n* = 12 sessions, 4 mice), and trials in control mice where blue light was delivered bilaterally to the cerebellum but PCs did not express ChR2 (*n* = 6 sessions, 2 mice). Asterisks indicate significant differences during photostimulation trials ($p = 0.0228$ and $0.0199$ for the unilateral and bilateral photostimulation conditions, respectively; ANOVA with Tukey's post-test). (**F**) Same as panel D but with the photostimulus timed to the period of earliest lick-bout initiations. Note the absence of licking during photostimulation and the large increase in licking immediately after photostimulation ended (*n* = 11 sessions, 3 mice). (**G**) Histogram of lick-bout initiation times for control and optogenetic stimulation trials (same data as panel F). A clear increase in licking probability is apparent after photostimulus ended. Only well-separated lick bouts were included (>2 s of prior nonlicking). (**H**) Summary of the effect of optogenetic PC activity perturbation on licking behavior during task performance. Asterisks indicate a significant reduction in licking during photostimulation (On: $p = 0.0079$ and $p = 0.0486$ for unilateral and bilateral stimuli, respectively; ANOVA with

*Figure 7 continued*

Tukey's correction for multiple comparisons) and a significant increase during the rebound period for bilateral stimulation (Rb: p = 0.0034, ANOVA with Tukey's correction for multiple comparisons). In the control condition, light stimuli were delivered to the cerebellum of non-ChR2-expressing animals (*n* = 12 sessions, 2 mice). See also *Figure 7—source data 1*.

The online version of this article includes the following source data and figure supplement(s) for figure 7:

**Source data 1.** Source data for *Figure 7B, E and H*.

**Figure supplement 1.** Unilateral optogenetic stimulation of Purkinje cells (PCs) during the task.

effect on licking in mice that did not express ChR2 (*Figure 7E*). Therefore, optogenetically disrupting PC activity during internally timed licking severely disrupts behavioral performance. This includes the parameters related to how PC activity in this region represents the movement, such as the timing of individual cycles of licking, which is important for establishing rhythmicity of closely spaced licks during a lick bout.

To further explore the ramifications of PC activity perturbation on motor-event transitions, we delivered optogenetic stimuli well before the time point of expected water allocation, when mice first began to elicit bouts of exploratory licking. During the PC photostimulation period, exploratory licking again largely abated (*Figure 7F*, *Figure 7—figure supplement 1*). However, immediately after the photostimulation, the mice initiated a barrage of licking at a probability much greater than that observed at the same time point in control trials (*Figure 7G*, *Figure 7—figure supplement 1*). Such optogenetically induced rebound licking was more prominent for bilateral photostimulation of Crus II PCs than for unilateral photostimulation of Crus I or II PCs and was not observed for light stimuli delivered to non-ChR2-expressing mice (*Figure 7H*). Together, these results indicate that in addition to perturbing cycles of tongue protraction and retraction, PC photostimulation can also both initiate and terminate bouts of licking, depending on the timing of the activity perturbation relative to the licking context, indicating a role for PC activity in coordinating the regularity of ongoing motor performance as well as motor-event transitions.

## Discussion

The temporal consistency of periodically performed discontinuous movement is believed to be aided by timing representations of salient motor events by the cerebellum (*Ivry et al., 2002*). In support of this idea, we observed changes in murine PC activity immediately prior to both the initiation and termination of licking bouts timed to regular intervals of water-reward allocation. Moreover, perturbation of this activity influenced the behavioral performance. Although the cerebellum has been implicated in learning temporal associations of predictive sensory cues and impending motor actions, the changes we observed in PC activity occurred while the animals performed well-timed, goal-directed licking independent of any overt sensory evidence indicating reward availability. Thus, our results indicate that this activity is internally driven and related to the timing of motor events that are pertinent to organizing the temporal consistency of volitional behavioral performance.

In rodents, consummatory licking is performed as a continuous rhythmic movement composed of repeated cycling of tongue protractions and retractions which is commanded by a brainstem central pattern generator (*Horowitz et al., 1977*; *Wiesenfeld et al., 1977*). Motor plans for volitional lick-bout initiation and the choice of directional licking are composed in the cerebral cortex, which depends on cerebellar input for accurate planning (*Gao et al., 2018*; *Li et al., 2015*). We found that intermingled PCs in the Crus I and II lobules displayed heterogeneous coding of licking-related attributes in their pattern of simple spiking suggesting that the cerebellum helps prepare and execute the resulting movement. PCs encoded parameters related to the phasic timing of individual licks that comprise bouts, similar to that observed in a prior study examining nonperiodically performed consummatory licking (*Bryant et al., 2010*), collectively mapping the entire spectrum of the lick cycle in their simple spiking activity. Climbing-fiber-induced activity in PCs has previously been shown to be aligned to licks in consummatory bouts, including in mouse Crus II (*Gaffield et al., 2016*; *Welsh et al., 1995*). The overall licking rate was also encoded in the ensemble PC simple spiking response. However, this effect was only observed across the entire spectrum of licking and was not apparent for subtle differences in lick rate behavior, for example, between the peak rate of exploratory and reactive licking bouts. These results indicate a rich, nested representation of multiple motor attributes occurring at different

timescales, reminiscent of PC activity during other rhythmic motor behaviors (*Sauerbrei et al., 2015*). In our recordings, we also readily observed activity ramping in individual PCs in advance of both lick-bout initiation and termination, suggesting a representation of impending motor-event transitions in this population. This activity appeared to be involved with the preparation of the change in vigor of the ensuing start and end of lick bouts; the former parameter informing the timing of the peak lick rate which, for our task, was temporally aligned to the predicted time of water-reward allocation.

Activity ramping in PCs has also been observed during other volitional motor behaviors. For example, in monkeys, PCs fire elevated barrages of simple spikes immediately prior to changes in eye-movement speed and/or direction during object tracking (*Herzfeld et al., 2015*) or arm movements during reaching behaviors (reviewed in *Ebner et al., 2011*). Increased simple spiking also briefly precedes spontaneous and sensory evoked whisking in some PCs of mice (*Brown and Raman, 2018*; *Chen et al., 2016*). However, compared to the close temporal correspondence between PC activity ramping and motor action in these earlier reports, lasting just a few tens of milliseconds, we found that the onset time of simple spike ramping preceded lick-bout initiation and termination by hundreds of milliseconds. This time frame is consistent with the ramping of cerebellar activity in advance of internally timed, volitional eye movements of monkeys that followed a delay interval of several seconds (*Ohmae et al., 2017*). Thus, the onset of PC activity ramping, relative to the ensuing movement, may undertake different dynamics dependent on the timing needs of the underlying behavior.

PC activity did not map the entire epoch of the delay period in our interval timing task, but rather only the end point immediately prior to lick-bout initiation. This result is consistent with the idea that multiple brain regions form timing representations of movements at different scales, with the cerebellum contributing largely to the subsecond range (*Tanaka et al., 2021*). Interestingly, we found that the onset time of PC activity ramping occurred earlier for licking that was preemptive, rather than reactive, to water-reward availability. Because reactive licking was likely triggered by an external cue, it may be that the shorter onset time of ramping in this context of licking promotes a more reflex-like response, perhaps emergent from sensorimotor associations formed in the cerebellum, ensuring that ensuing consummatory movements are executed with little delay. By contrast, when the timing representation of the impending motor event is internally rather than externally signaled, such as for exploratory licking elicited prior to water allocation, PC activity may ramp earlier due to input (either direct or indirect) from a separate brain region that also participates in organizing the behavior. Contributing brain regions could include the basal ganglia, which also forms a time representation of discontinuous movement, but with a longer timescale than the cerebellum (*Kunimatsu et al., 2018*; *Ohmae et al., 2017*). In mice, optogenetic stimulation of an inhibitory basal ganglia pathway catastrophically disrupts licking during the same interval timing task, indicating that this region also plays a role in organizing the behavior (*Toda et al., 2017*). The cerebellum is also recurrently connected with the cerebral cortex, forming a loop whose activity helps maintain motor plans in working memory until initiation (*Gao et al., 2018*; *Svoboda and Li, 2018*). Because mice must carefully track the passage of time to correctly anticipate their next planned cycle of licking around water-reward allocation, earlier PC activity ramping during exploratory licking bouts may also reflect the engagement of cortical circuitry.

The termination of motor action is also a salient event that delimits discontinuous movement. However, in comparison to movement initiation, less is known about how cerebellar activity at the end of each movement cycle encodes and influences periodically performed behaviors. We observed widespread ramping of simple spiking in individual Crus I and II PCs at lick-bout termination, although the population response was relatively weak, especially for unrewarded trials. During task training, the mice learned to adjust their licking rate, ultimately stopping after consuming the dispensed water droplet on rewarded trials. Thus, the diminishing quantity of water during consumption may signal an impending need to end the action based on prior sensorimotor associations formed during task training. Mice also stopped licking on most unrewarded trials, presumably because the estimated time window for the expected water reward had passed. For unrewarded trials, there was no sensory information to cue an impending end to action. After giving up, the mice began waiting for the next period of reward allocation to elicit their planned behavior. Thus, we speculate that the uptick in PC simple spiking toward the end of licking bouts helps prepare an impending stop to motor action, similar to the modulation of cerebellar activity at the end of dexterous reaching behaviors that influences kinematics and endpoint precision of grasps (*Becker and Person, 2019*; *Low et al., 2018*). We

propose that this activity may be necessary to coordinate precise temporal control of the subsequent cycle of movement.

Although we observed abundant positively modulating PCs in Crus I and II during periodically performed bouts of licking, we recorded relatively few PCs with negatively modulating simple spike responses. This result is surprising because cerebellar-dependent motor control has been attributed to increases in output from the cerebellar nuclei, which is expected to result from the absence of PC-mediated inhibition (*Ten Brinke et al., 2017*). However, the influence of PCs on movement is believed to manifest at the population level, resulting from the cerebellar nuclei integrating activity from both positively and negatively modulating PCs (*Calame et al., 2021*; *Herzfeld et al., 2015*). It also remains possible that the activity we observed is predominately from PCs that play a supporting role in movement organization, participating as an antagonistic module that is recurrently connected with a direct, motor-driving module whose PCs oppositely modulate their simple spike firing pattern (*Ohmae et al., 2021*). Further work, using anatomical and functional tools to delineate the targets of both positively and negatively modulating PCs, as well as the ability to independently toggle their activity, will be necessary to address this question.

In monkey Crus I and II, PCs encode movement-related activity, but not reward or reward-expectation activity (at least in the absence of learning), in their simple spiking during the performance of a reward-driven motor task (*Sendhilnathan et al., 2020*). While our results point to a likewise representation of motor-timing events in Crus I and II PC simple spiking, we cannot fully discount the possibility that expectation information is also encoded in the same PC population, as has been shown for cerebellar granule cells (*Wagner et al., 2017*). Because preparatory neural activity is inexplicably linked to expectation when volitional movements are performed to acquire rewards, it may be that the predictively timed licking behavior we observed may have benefited from such information. For example, the cerebellum may have harnessed the sensory feedback of water availability to guide temporal learning so that licking was periodically elicited around reward availability and ended when the dispensed water was fully consumed.

Climbing fibers provide instructive signals to guide supervised or reinforcement learning (*Hull, 2020*; *Raymond and Medina, 2018*). Interestingly, in addition to the ramping of PC simple spiking, we observed an increase in climbing-fiber-evoked activity in a PC subpopulation, which preceded lick-bout initiation by ~100 ms. Although climbing fibers are responsive to sensory stimuli (*Gaffield et al., 2019*; *Ohmae and Medina, 2015*), the activity increase occurred in advance of exploratory licking bouts that were initiated without an overt sensory cue, ruling out the possibility that this activity represented an external stimulus. Climbing-fiber-evoked activity did not change at lick-bout termination, suggesting a specific role for this input only at the start of each movement cycle. Reward-related climbing fiber signaling is prevalent in the cerebellar cortex (*Heffley and Hull, 2019*; *Heffley et al., 2018*; *Kostadinov et al., 2019*); thus, the dramatic increase in climbing-fiber-evoked activity in PCs during behavior could reflect a prospective response to an anticipated reward. However, the same PCs were unresponsive in water-omission trials, indicating a lack of apparent reward prediction errors. Therefore, in our view, this movement-aligned activity is more likely motor related.

Because the synchronization of climbing-fiber-evoked complex spiking within parasagittal-aligned clusters of PCs can evoke and/or invigorate motor action, including bouts of sensory-triggered licking (*Apps and Hawkes, 2009*; *Ten Brinke et al., 2017*; *Tsutsumi et al., 2020*; *Welsh, 2002*), the activity increase we observed in PCs due to climbing fiber input could influence the kinematics of lick-bout initiation if this activity were temporally correlated in the PC population. However, as we used small fields of view during our optical recordings, there were generally too few simultaneously active PC dendrites to accurately quantify the level of their synchrony. For this reason, we were unable to determine whether the responsive and unresponsive regions of Crus I and II during task performance corresponded to distinct, functional clusters of PCs that have been shown to be either engaged or not engaged during sensory-driven licking (*Tsutsumi et al., 2020*).

Our optogenetic experiment provided causal evidence that Crus I and II PC activity influences movement performance. For example, licking rhythmicity was degraded by PC photostimulation suggesting that the cerebellum is capable of modulating the central pattern generator. PC photostimulation also elicited motor-event transitions resembling those occurring during planned, periodically performed licking. Perturbating PC activity terminated ongoing licking, even when the stimulus was timed to the peak output rate around expected water rewards. This perturbation could also trigger

lick-bout initiations when the photostimulation ended. However, optogenetically evoked licking was apparent only when the perturbation was timed to the period when the animals were beginning to elicit exploratory bouts of licking in anticipation of water rewards. Therefore, like the susceptibility of the cerebellum to instantiate learning (*Albergaria et al., 2018*), our results indicate that there may be a conditional state during which the cerebellum is more effective at triggering movement initiation, in particular when planned movements are first being converted into motivated actions, which may require a differential engagement of the cerebellar–thalamocortical pathway compared to online control of the ensuing movement (*Nashef et al., 2021*). These behavioral effects are consistent with prior reports. For example, a study found that optogenetic perturbation of cerebellar activity reduces voluntary whisker movements during the photostimulation period and leads to a subsequent rebound in whisking behavior (*Proville et al., 2014*). Because PCs form a nested representation of multiple behavioral attributes related to periodically performed discontinuous motor behavior, it may that the effects of the PC optogenetic perturbation during licking, or at lick initiation, may stem from disruption of signals related to either individual tongue protrusions and retractions and/or lick-bout initiation or termination. One way to fully disambiguate these effects in the future would be to separately perturb CPG-targeting and non-CPG-targeting PCs, if such pathways exist. Furthermore, we did not assess the circuit effect of autonomous PC photostimulation during behavior. Therefore, we cannot draw any conclusions between the precise pattern of induced activity and motor-event outcomes. However, PC photostimulation produces both direct increases and indirect decreases in simple spiking in the PC ensemble and can drive complex-spike-like bursts of activity (*Bonnan et al., 2021*; *Tsutsumi et al., 2020*) which all may be important for motor preparation and execution.

In summary, PC activity encodes and influences cycles of repeat actions and both represents and causes motor-event transitions. Thus, the coordination of explicitly timed, volitional movements is improved by the cerebellum, consequently altering their temporal consistency across repeat cycles of goal-directed action.

## Materials and methods

### Animals

All animal procedures were performed following protocols approved by the Institutional Animal Care and Use Committee at the Max Planck Florida Institute for Neuroscience. Adult animals (>10 weeks) from the following strains of mice were used in this study: C57/Bl6 (5 f, 3 m; Jackson Lab stock: 000664), $Pcp2^{Cre}$ (1 f, 2 m; Jackson Lab stock: 010536), $Pcp2^{Cre}$ crossed with Ai27 $Gt(ROSA)26Sor^{tm27.1(CAG-COP4*H134R/tdTomato)}$ (5 f, 8 m; Jackson Lab stock: 012567), and nNOS-ChR2 (1 f, 2 m) in which ChR2(H134R)-YFP expression is controlled by the *Nos1* promoter (*Kim et al., 2014*; *Madisen et al., 2012*; *Zhang et al., 2004*). These mice were maintained on a 12-hr light–dark cycle with ad libitum access to food and were provided a running wheel for enrichment.

### Surgical procedures

For all surgeries, we used isoflurane for anesthesia (1.5–2%). A heating pad with biofeedback control provided body temperature maintenance. Buprenorphine (0.35 mg/kg subcutaneous), carprofen (5 mg/kg subcutaneous), and a lidocaine/bupivacaine cocktail (topical) were used for pain control. To restrain the head during behavioral experiments, a small stainless-steel post was attached to the skull. To facilitate neural activity recording, we performed a craniotomy (~2 mm square) over the left lateral cerebellum, above a region that included portions of the Crus I and II lobules (*Gaffield et al., 2016*). A glass coverslip was cemented over this area to protect the brain postsurgery while the animals were trained for the behavior. For some optogenetic experiments, optical fibers (MFC_400/430–0.48_MF1.25_FLT, Doric Lenses, Quebec, Canada) were instead installed bilaterally over Crus II (3.5 mm lateral, 2.2 mm caudal from lambda). All implants were fixed in position using Metabond (Parkell, Edgewood, NY). For calcium imaging experiments, a subset of mice were injected with adeno-associated virus (AAV)1 containing the genetically encoded calcium indicator GCaMP6f (*Chen et al., 2013*) under control of the Pcp2 promoter (*Nitta et al., 2017*) at the brain area under the craniotomy to transduce Crus I and II PCs. All mice were given at least 7 days to recover from surgery before beginning behavioral training and/or experimentation.

## Behavioral procedures

For the interval timing task, mice were held under head-fixation in a custom-built apparatus consisting of a metal tube (25.4 mm diameter) in which the mice rested comfortably and a metal water port that was placed in front of their mouths. This port was calibrated to allocate 4 µl of water per dispensed droplet as determined by the open time of a solenoid valve (INKA2424212H, Lee Company, Westbrook, CT). This apparatus was housed inside a sound-insulated and light-protected enclosure. The water valve was located outside of the enclosure to prevent the mice from hearing it open and close. Licks were detected using a simple transistor-based lick circuit connecting the metal tube to the metal water port. This circuit closed when the tongues of the mice contacted the water port. The apparatus was controlled by a BPod state machine (Sanworks, Rochester, NY) installed on a Teensy 3.6 microcontroller (SparkFun Electronics, Boulder, CO) combined with custom-written codes (Matlab, MathWorks, Natick, MA).

Thirst was used to motivate behavioral performance. To achieve this, mice underwent water restriction with daily water intake limited to 1 ml with frequent monitoring to confirm the lack of any adverse health consequences (*Guo et al., 2014*). Initial sessions of behavioral training consisted of a block of 50 consecutive water-rewarded trials to reinforce the target time interval. Thereafter, sessions consisted of a trial structure compromising 80% rewarded and 20% unrewarded trials that were randomly distributed (although consecutive unrewarded trials were prevented). Mice typically completed 250–300 trials per session, with only a single session per day, and were considered well trained when they consistently initiated licking bouts prior to water delivery, and they accurately anticipated the water delivery time on unrewarded trials (peak lick rate within 0.5 s of the expected time of water allocation). Most mice reached this criterion level after 10–15 training sessions.

## Electrophysiology

The cerebellum was accessed for electrophysiology recording through the previously prepared craniotomy that exposed large portions of both the Crus I and II lobules including zebrin bands 7+, 6−, and 6+. On the day of recording, and under light isoflurane anesthesia, the coverslip was removed by removing the Metabond that secured it in place. A silver wire was then place into the craniotomy site to provide a ground signal. At least 45 min after recovery from anesthesia, the mouse was transferred to the behavioral apparatus and the brain and ground wire were covered with a saline solution. A silicon probe (A1 × 32-Poly3-5mm-25s-177, Neuronexus, Ann Arbor, MI) was then slowly inserted into the exposed cerebellar cortex, under view of a high-magnification mini-camera, at a few microns per second using a motorized micromanipulator (Mini, Luigs and Neumann, Ratingen, Germany). Because of the large size of the craniotomy, both Crus I and II could be visually distinguished in the video image, allowing the targeted localization of the probe recording site to either lobule of interest by investigator choice. The target depth was approximately <500 µm. The silicon probe was connected to an amplifier (RHD2132, Intan Technologies, Los Angeles, CA) and read out by a controller interface (RHD200, Intan Technologies) with a sampling rate of 20 kHz. Commercial software (Intan Technologies) was used for data acquisition. For electrophysiology experiments combining optogenetics, laser light was delivered directly onto the exposed brain using a patch cable that was carefully positioned to minimize any induced artifacts in the recordings while still targeting the targeted region of the cerebellar cortex. A copy of the electrical signal driving the light stimulus was recorded to ensure proper registration to the electrophysiological and behavioral recordings.

## Calcium imaging

All two-photon imaging experiments were performed as previously described (*Gaffield et al., 2016*; *Gaffield et al., 2019*). Briefly, a custom-built, movable-objective microscope acquired continuous images at ~30 frame/s using an 8 kHz resonant scan mirror in combination with a galvanometer mirror (Cambridge Technologies, Bedford, MA). A ×16, 0.8 NA water immersion objective (Olympus, Tokyo, Japan) was used for light focusing and collection. The lens was dipped in an immersion media of diluted ultrasound gel (1:10 with distilled water) that was applied to the cranial window the day of recording. The microscope was controlled using ScanImage software (Vidrio Technologies, Ashburn, VA). GCaMP6f was excited with pulsed infrared (900 nm) light from a Chameleon Vision S laser (Coherent, Santa Clara, CA) with an output power of <30 mW from the objective.

## Optogenetics

In optogenetic experiments, stimulation of ChR2 was driven by a 473 nm continuous-wave laser (MBL-F-473-200 mW; CNI Optoelectronics, Changchun, China). An acousto-optical modulator (MTS110-A3-VIS controlled by a MODA110 Fixed Frequency Driver; AA Opto-Electronic, Orsay, France) modulated the laser power to produce brief pulses of light during experimental procedures (40 Hz, 5 ms). For this, the laser light was directed into a patch cable (BFYL2F01; Thorlabs, Newton, NJ) that was either directly placed over the installed cranial window (unilateral stimulation) or connected to the implanted optical fibers (bilateral stimulation). For unilateral stimulation of either Crus I or II, the output of the patch cable was set to 15 mW and the surrounding area was covered with black foam to limit visibility to the mouse. For bilateral stimulation experiments, the light output was ~2.75 mW; the ceramic connectors (ADAL1; Thorlabs) were covered in black heat shrink tubing, and then covered again with small black tubes to limit light visibility to the mouse. All optogenetic experiments included an overhead 470 nm light emitting diode (M470L4; Thorlabs) that was continuously directed at the mice to mask the light flashes used for optogenetic stimulation.

## Data analysis

Licking behavior was analyzed by calculating the lick rate as the inverse of the interlick interval. To quantify correlations between lick rate and the simple spiking rate in PCs, the mean lick rate for each trial was sorted into 1 s bins. Only trials in which the first lick occurred at least 2 s into the trial were included to ensure enough prelick electrophysiological data for analysis. Similarly, only trials in which the last lick had at least 1 s of postlick electrophysiological data were included in the analysis. In many analyses, we only focused on well-isolated licking, defined as bouts without preceding licking for 2 s. For optogenetic perturbation experiments, the lick variance was normalized to the prestimulus control condition for each session.

Silicon probe data were sorted automatically by the Kilosort algorithm (*Pachitariu et al., 2016*), followed by manual curation using Phy2 software (https://github.com/cortex-lab/phy, *Rossant, 2021*) into 202 unique clusters. In a first-pass analysis, PC units (*n* = 48) were unambiguously identified by some accompanying complex spikes in the recordings (*Figure 2—figure supplement 1A*). We also confirmed that these units had the simple spiking properties expected for PCs (i.e., firing rate and regularity) (*Van Dijck et al., 2013*). There were an additional 56 units with similar simple spiking responses, but complex spikes could not be clearly discerned in their activity recordings. Noticeably, spiking in unambiguously identified PC units was clearly represented in many nearby channels of the silicon probes (*Figure 2—figure supplement 1C*). Therefore, we calculated the mean peak spike size in each channel and counted the number of channels with a peak significantly above the noise (~30 µV). This approach determined that the channel count for unambiguously identified PCs tended to be quite high (i.e., ≥7), likely due to the large size of PCs relative to the spacing of the electrode pads on the silicon probes (*Figure 2—figure supplement 1C,D*).

To test whether channel count representation could be used to classify PCs, we compared the spike distribution of unambiguously identified PCs in each channel with those of another abundant cerebellar cell type, molecular layer interneurons (MLIs). We did not choose to examine granule cells, another abundant cell type, in this comparison because granule cells typically have very different behavior-evoked firing properties than PCs (*Powell et al., 2015*). In addition, due to the low impedance of the electrode pads used on silicon probes, the activity of granule cells is generally not detectable in extracellular electrophysiological recordings. To positively identify MLI units in our recordings, we used an opto-tag strategy whereby ChR2-expressing MLIs in nNOS-ChR2 mice (*Kim et al., 2014*) had short-latency responses to light pulses (*Figure 2—figure supplement 1B*; recordings were not obtained from these mice during behavior). In comparison to unambiguously identified PC units, spiking in identified MLI units was detected in only a few channels (*Figure 2—figure supplement 1C-E*). Therefore, a high representation of spiking activity in many channels (≥7) could successfully classify PCs. Using this criterion to identify additional putative PCs, we included another 41 units, from the 56 showing similar spiking response characteristics, for a total of 89 in our analysis.

When quantifying the entrainment of PC spiking to the lick cycle, we only included licks that occurred in sequences of at least three consecutive licks with interlick intervals of 100–175 ms. For each PC, the phase of the lick cycle at which each spike occurred was obtained by linear interpolation between consecutive licks. Tongue contact with the sensor was defined to be 0 = 2$\pi$ radians. To

determine whether each PC was entrained to the licking rhythm, a Rayleigh test was performed. To control for multiple comparisons across neurons, a Benjamini–Hochberg correction was applied with a false discovery rate of 0.05. The distribution of lick phases at spike times was visualized using histograms with a bin width of $\pi/10$. The circular density of lick phases was computed using a kernel density estimator with a kernel width of 0.3. The phase and magnitude of entrainment were characterized using the angle and length of the mean resultant, respectively.

We manually examined the data to classify PCs into groups based on how their simple spiking activity was tuned to different types of motor-event transitions during the behavioral task. PCs were classified as activated during the first lick in a bout if their simple spiking rate showed three consecutive increases in 100ms time bins that began 1 s before that first lick and ended 0.5 s after that same first lick. The ramp onset time was the time bin of the first of those rate increases. PCs were classified as activated during the last lick in a bout based on the mean simple spike rate during the 300 ms preceding the last lick. Negatively modulated PCs were classified using the same criterion except we used negative values instead. In some cases, individual PC recordings were noisy enough that our rigid classification may have missed some responding cells, so we expect our results to represent a lower-bound estimate of tuning specificity. To quantify population activity, we used the change in simple spike rate ($\Delta$spike rate). For this metric, we determined the average firing rate for all nonlicking periods for each PC during the recording and used this level of activity as the baseline from which movement-related firing was computed for that cell. For the last lick analysis, $\Delta$spike rate was alternatively calculated relative to PC activity during the 1–5 s preceding the last lick.

Calcium imaging data were analyzed using a standard procedure (*Gaffield et al., 2016*). In brief, individual PC dendrites were identified using independent component analysis (*Hyvärinen, 1999*). Calcium events were then extracted from the raw fluorescence traces using an inference algorithm (*Vogelstein et al., 2010*). Regions of interest (ROIs) in the imaged areas of Crus I and II were considered water responsive if a peak in activity was observed in the total PC average for that region at the time of water delivery. The ramp onset time was determined when the peak event rate reached >3 standard deviations above the mean, in 100-ms time bins, beginning 1 s before and up to 0.5 s after the first lick in a lick bout. To quantify calcium event activity prior to the last lick, we used a similar criterion, but examined event rate activity at least 2-s post water delivery in rewarded trials as well as at the end of licking in water-omission trials.

In the figures, shaded areas in activity plots indicate standard error of the mean (SEM range); error bars are also represented as SEM. Statistical values were calculated using GraphPad (Prism, San Diego, CA) with significance indicated by p values below 0.05.

## Acknowledgements

We thank Samantha Amat for laboratory assistance and the GENIE program (Janelia Research Campus, including Drs. Jayaraman, Kerr, Kim, Looger, and Svoboda) for freely providing GCaMP6f to the neuroscience community.

## Additional information

### Funding

| Funder | Grant reference number | Author |
|---|---|---|
| National Institute of Neurological Disorders and Stroke | NS1188401 | Jason M Christie |
| National Institute of Neurological Disorders and Stroke | NS105958 | Jason M Christie |
| National Institute of Neurological Disorders and Stroke | NS112289 | Jason M Christie |

| Funder | Grant reference number | Author |
|---|---|---|
| Max Planck Florida Institute for Neuroscience | open access funding | Michael A Gaffield Jason M Christie |

The funders had no role in study design, data collection, and interpretation, or the decision to submit the work for publication.

## Author contributions
Michael A Gaffield, Conceptualization, Investigation, Writing – original draft, Writing – review and editing; Britton A Sauerbrei, Formal analysis, Writing – review and editing; Jason M Christie, Conceptualization, Funding acquisition, Project administration, Supervision, Writing – original draft, Writing – review and editing

## Author ORCIDs
Britton A Sauerbrei (iD) http://orcid.org/0000-0003-3386-3243
Jason M Christie (iD) http://orcid.org/0000-0003-0276-2554

## Ethics
All of the animals were handled according to approved Institutional Animal Care and Use Committee (IACUC) protocols of the Max Planck Florida Institute for Neuroscience (Protocol Number: 18-009) . As detailed in Methods and materials, care was taken to minimize animal pain, suffering, and distress.

## Decision letter and Author response
Decision letter https://doi.org/10.7554/eLife.71464.sa1
Author response https://doi.org/10.7554/eLife.71464.sa2

# Additional files

## Supplementary files
• Transparent reporting form

## Data availability
All data generated or analyzed during this study are included in the manuscript and supporting files. Source data files have been provided for Figures 4, 6 and 7.

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
