## [Editor Report]

This study conducts physiological recordings in awake mice to reveal how cerebellar Purkinje cells convey temporal information about the onset and offset of ongoing movements. There is a growing appreciation for the cerebellum in planned behavior, but how it contributes to a spontaneously initiated volitional behavior remained unclear. This work provides important insights into the roles of cerebellar Purkinje cells in regulating rhythmic movements under volitional control.

---

## [Decision Letter]

**Decision letter after peer review:**

Thank you for submitting your article "The cerebellum encodes and influences the initiation and termination of discontinuous movements" for consideration by *eLife*. Your article has been reviewed by 3 peer reviewers, one of whom is a member of our Board of Reviewing Editors, and the evaluation has been overseen by Tirin Moore as the Senior Editor. The following individual involved in review of your submission has agreed to reveal their identity: Shogo Ohmae (Reviewer #2).

The reviewers have discussed their reviews with one another and agreed on the potential importance of the study. However, the reviewers have raised some critical concerns which need to be clarified by additional experiments and/or analysis. The Reviewing Editor has highlighted these concerns, and the comments from three reviews are listed below.

Essential revisions:

1) The authors show that the simple spikes but not complex spikes ramp up before the initiation and termination of lick bouts. While this finding is important, the causality between the simple spikes and motor initiation/termination is unclear. In general, simple spikes fire more when the lick rate increases and confounds the interpretation of ramping activity (Reviewer 2). Furthermore, the trained licking behavior may not be well-timed enough to obtain a clear correlation between neuronal activity and behavior (Reviewer 1). These issues need to be clarified either by additional experiments or analysis.

2) The results from the optogenetic experiments did not sufficiently support the main conclusion. Because the light stimuli by itself caused significant changes in licking, the shift in the lick–peak may have been induced indirectly by this abnormal licking behavior and not directly by the changes in simple spike activity (Reviewer 1-1, Reviewer 2-4). In addition, because optogenetic stimuli did not specifically change simple spikes, the involvement of complex spikes cannot be ruled out (Reviewer 3).

*Reviewer #1 (Recommendations for the authors):*

Specific concerns:

1) It is difficult to interpret the optogenetic perturbation experiments in absence of corresponding neural data. While recording during such experiments certainly imposes an additional experimental burden, it is necessary to understand what essential features of PC activity dictate behavioral performance. In particular, it is striking that the same stimulation only produces a delay in peak licking when licks are initiated less than one second from the stimulus. Corresponding neural data revealing what differs between trials with short and long latency lick initiations would enable a much stronger interpretation of this result.

Related to this concern, Figure 7 demonstrates that optogenetic stimulation initiates licks on some (but not all) trials immediately following stimulation. The spike above baseline in lick initiation probability (Figure 7G) occurs < 1 s following optogenetic stimulation. This may suggest that the delay in peak lick rate only occurs when non–voluntary licks are initiated too close to the target time (i.e. on the trials where cessation of optogenetic stimulation drives immediate rebound licking)? The sub vs supra–second timing result could therefore relate to 1) the animals' voluntary initiation of licking too close to the optogenetic perturbation on a subset of trials, or 2) the animals' immediate involuntary response to cessation of optogenetic stimulation on a subset of trials. These two possibilities may suggest different interpretations for how the cerebellum contributes to the behavior.

2) It is not clear that the observed responses relate to motor initiation and termination. For example, considering figure 4C vs 4E, the Sspk ramping at lick offset is quite different for omission trials as compared to rewarded trials. If these signals primarily reflect a timing response for stopping the motor program, they should not depend so heavily on reward context. Alternatively, the initiation signals could also be interpreted in the context of expectation. This is very challenging to disentangle, however. In this behavior, the omission trials do not serve well to do so, as it is not possible to align the data appropriately. Figure 2C shows that the animals have a poor temporal estimation of reward delivery on average, and maintain near–peak licking across an approximate 5 second window in absence of reward. Thus, because animals do not have a strong estimation of when the reward will occur, alignment to 'expected reward time' on omission trials is unlikely to capture any underlying responses. Related to this, given that Wagner et al. showed that granule cells also have reward omission responses in Crus I – it is important to reconcile this difference, even if it is simply because omission responses may not be identifiable given this task design.

Also related to this concern, the discussion argues that "our results reinforce the idea that the cerebellum influences well–timed, regularly performed actions that are generated solely by motivation...". This may not be accurate, and illustrates a major confound in interpreting signals during this behavior. Certainly, naïve animals are driven solely by motivation. They do not, however, exhibit any predictively timed licking behavior. In contrast, trained animals have harnessed the reward–related sensory stimulus to generate a new predictive behavior. This difference between naïve and trained animals shows that reward does double duty in this task – it is both a sensory stimulus that has intrinsic positive valence (and is thus sought solely by motivation in naïve animals), and a sensory cue that can guide temporal learning. Because naïve animals exhibit the former but not the latter, these two roles are separable. However, this dual role of the sensory input in this task also makes the interpretation of the neural responses in the learned condition challenging. In other words, because the cue and reward are one in the same in this task, it is extremely difficult to disambiguate sensory, expectation/prediction, and motor related signals. I am concerned that the task design may preclude causal relationships between the neural activity and behavior as a result.

Additional questions:

1) What is the depth of silicon probe recordings (what is the range of depths of the primary contacts for recorded PCs?). Does this overlap with imaging depths, or does the imaging data come from different regions than the electrophysiology? Do the Cspks recorded with electrophysiology exhibit the same behavior as those captured with calcium imaging?

2) What is the definition of a lick bout? Why is the average absolute lick rate (in s–1) pinned at 0 with no error for 2 seconds prior to lick bout initiation (e.g. Figure 3A)? It seems that bout initiations begin about 2.5 seconds prior to reward (Figure 3A bottom). Figures 1 and 2 show continuous licking on average for 5 seconds prior to reward. Doesn't this mean that there should be some non–zero lick rate prior to bout initiation? Alternatively, if these are licks defined by 2 seconds of preceding quiescence, shouldn't the initiations in the bottom of figure 2A bottom be further from the reward time?

3) It looks like there is a Cspk response to the last lick in figure 6I – is this significant? Is this response different between rewarded and reward–omission trials?

4) How are lick initiations defined surrounding optogenetic stimulation? Optogenetic stimulation suppresses licking, and results in rebound responses afterwards. When these occur in the pre–water period, they are defined as initiations. Why aren't the post water licks that follow the quiescence period of optogenetic stimulation considered initiations? Would not the same analysis show in 7G (related to 7F) yield a similar spike in initiation probability if performed on the data from 7D?

5) It's a bit odd to argue that imaging was necessary because it is "challenging to reliably distinguish complex spike waveforms" (line 337), while at the same time making the case that cell sorting was validated based on the property that PC units were 'unambiguously identified by accompanying complex spikes in the recordings" (line 774).

*Reviewer #2 (Recommendations for the authors):*

I recommend the following analysis, presentation, and discussion to improve the paper.

For the first major comment, the authors need to evaluate the encoding of each bout of licking by cross–correlation or event triggered averaging, and separate it from the encoding of initiation and termination (e.g. using linear model).

For the second comment, they should further analyze the negatively modulating Purkinje cells, shown in Figure 5C. They also need to display the histology of their recording sites (Schematic drawing in Figure 2A is not enough). Also, they should discuss about the potential bias in recording of Purkinje cells.

For the third comment, I recommend the spike rete–based display for the data presentations, which should include the spontaneous spike rates.

For the fourth and fifth comments, related to the first comment, if the Purkinje cells contribute to controlling each of licking bout, the observation in the optogenetic experiments could be explained. The authors need to consider the possibility and improve the interpretation and conclusion.

*Reviewer #3 (Recommendations for the authors):*

– Figure 3 E shows the difference in simple spike onset between pre and post licking. It seems both Crus I and II were pooled together, but this analysis should be conducted separately in different lobules.

– Figure 3 F shows representative PC activity relative to the first lick. Is this trace from pre–water lick or post–water lick? Representative traces from both pre–water and post–water lick should be presented to demonstrate the difference in the ramp onset.

– Figure 4C shows spike rate changes before the termination of the lick. As the firing rate in Figure 3C–D is plotted in (s–1), this data should be also presented using the same unit for comparison.

– Figure 5D, please revise color coding for easier interpretation.

– Figure 5D shows a significant number of cells in Crus II ramped up for both pre–water and post-water licking. It would be informative to compare the ramp onset between pre-water and post–water licking for these cells. The result will show whether the same cells responded differently to the self–initiated and externally triggered licking.

– In Figures 7 and 8, optogenetic stimulation of PCs was conducted with ChR2. While this experiment was performed to test the involvement of simple spikes, it could be mediated by alteration in complex spikes. Were complex spikes affected by the stimuli? Representative traces from raw data for Figure 7A should be presented to address this point.

– Figures 8B and 8F show the delayed peak in lick rate with PC activation. This finding should be supported with additional statistical analysis.

[Editors' note: further revisions were suggested prior to acceptance, as described below.]

Thank you for resubmitting your work entitled "The cerebellum encodes and influences the initiation and termination of discontinuous movements" for further consideration by *eLife*.

We have consulted the original reviewers and have agreed that the revised version improved significantly. However, several issues remain and need to be clarified. Please read the reviewers' comments below and try to address them fully. Please note that points #1 and #2 in Reviewer 1's comment are critical for a successful revision: both reviewers agreed that these are the major weaknesses of the paper.

*Reviewer 1:*

I have several comments below directly related to our initial reviews. I think the paper is interesting, and that they could revise one more time with no experiments to address these issues.

The main concerns about the initial submission focused largely on the question of whether the authors' data and analysis support the central claim that the measured neural activity patterns are causally related to motor initiation and termination. In response to these concerns, the authors have provided several important clarifications and some new analyses. While I remain positive about the potential impact of the findings in this study, and would support this publication for *eLife*, there are some important remaining issues directly related to those central concerns.

1. Now that the authors have clarified the analysis related to figure 4, I am concerned about its main conclusions. Specifically, the authors state that "the average simple spiking rate began to ramp earlier for exploratory licking trials, when the movements were initiated prior to water allocation, compared with that for trials in which consummatory licking commenced immediately after water became available" (lines 205–208).

This is a key point meant to distinguish neural responses on planned vs unplanned movements, but I am not convinced by the underlying analysis. The authors have subtracted z–scored neural data for pre and post water licks, and used the difference to suggest that pre water licks have an associated neural responses that "starts earlier". However, this analysis conflates response amplitude and timing, as a subtraction only reveals when responses diverge (the negative latency in the difference trace cannot be taken to mean that the larger response began earlier). To specifically address timing (and not amplitude), the responses must first be scaled, and then subtracted. Because a smaller response takes longer to cross the same amplitude threshold, a subtraction will show a difference right away, even if the responses start at the same time. In addition, the quantification as performed in Figure 4F will necessarily indicate shorter latencies for larger responses.

It seems very likely that a scaled subtraction will show no latency difference for the data in figure 4C and D, and this would contradict the conclusion of a timing difference in neural responses for planned and unplanned movement.

The data in Figure 4 do, however, show a much smaller response on unplanned movements, which is interesting. However, it is unclear whether or not the licking is different in this case? This needs to be shown for the associated neural traces in figure 4. The overall data suggest that lick bouts are relatively homogenous at their initiation. Thus, these data may indicate that the amplitude of the neural response does not reflect lick rate, and the timing of the response does not relate to prepared or unprepared movement. As reviewer 2 notes, PC activity also does not seem to represent peak lick rate timing.

Together with the above, I therefore remain unclear on the authors model linking neural activity and behavior, and whether the recorded activity relates to motor preparation / planning / execution of some kind.

Other important issues:

2. In reading the other reviewers concerns about z–scoring and the authors responses, I realized that the z–scoring is performed to different baselines for different analyses, and not to the mean spike rate across the trial or a common reference period across analyses (the specifics of the z–scoring are not in the methods). This can be seen in difference between figures 4C and 5C, for example. I am concerned about this practice for a couple of reasons:

2.1) it means that the amplitude of neural signals cannot be compared across different figures and conditions. This makes it challenging to interpret the relationship between firing changes and behavior.

2.2) to disambiguate other explanations for the neural data in this paper, it would be helpful to further leverage conditions where behavior differs. For example, in Figure 4C, there are trials where no licks occur until after presentation of water. This affords the opportunity to ask whether or not there was any increase in spike rate before water allocation in absence of licking (and thus test whether changes in spike rate might reflect something other than motor initiation). However, the authors have z–scored to the mean immediately preceding post–water licks, which would obscure any such changes (Figure 4 supp. 2 may show such an increase pre–water?). While I am sympathetic to the authors' arguments that z–scoring is commonplace for neural recording data, the implementation here is not ideal for evaluating the relationship between spiking and behavior.

3. In the first round of reviews, the question of how to appropriately disambiguate expectation based on omission trials was raised, given the animals imprecise expectation of the time of water delivery. This concern necessitates a more convincing analysis in order to support the authors statements in the discussion regarding expectation signals. For example, with complex spiking on omission trials, alignment to the first lick after the time of water expectation would provide a more appropriate timepoint to indicate the animals' expectation. Even better would be to look at the moment when lick bouts start to decrease / are terminated on omission trials, as this is the timepoint when the animals' behavior indicates recognition that the expected reward is not present (and this time has previously been shown to reveal such expectation signals, at least in some conditions). There may indeed be no evidence of expectation signals in this behavior, but in absence of such analysis to evaluate the question appropriately, it seems premature to make such conclusions.

4. This point is only a suggestion, but I think others may have the same confusion regarding figure 8. The difference between 5 and 10 second trials is not that licking ramps up more slowly on 10 second trials – rather, the mean lick plots shown here speak to the probability of lick bout initiation across trials. On single trials, licking just goes from nothing to the patterned bout rate. On average, however, this manifests as a ramp due to variability in the onset of lick bout times, and the increase in probability of initiation as the trial progresses. The essential feature of 10 second trials is that lick bouts start on average later than they do for 5 second trials. This necessarily means that lick bouts are more likely to be outside of the 1.25 second window from optogenetic stimulation defined by the analysis in 8F. However, if the same analysis from figure 8E and F were performed for 10 second trials (<1 vs > 1.25 second initiations from stimulation), it should yield the same result. It is likely that the average licking for 10 second trials following stimulation only looks different because it is weighted so much more heavily toward bouts initiated >1.25 seconds from stimulation.

This analysis would greatly enhance the authors point that these effects are all about the duration from optogenetic stimulation when the animal tries to lick, and not about the absolute duration of the trial (as many will likely assume based on the figure and its description). Perhaps this seems obvious, but such clarification could provide strong support for the authors arguments related to the final figure.*Reviewer 2:*

The paper contains very interesting topics and findings and has a potential impact worthy of *eLife*, but I think it is still too immature to be published with minor revisions. I think the opinion of revising it again without experiments is very valid.

I also agree with the concerns of Reviewer #1. In particular, I have exactly the same concerns about the main concern and #2 issue, and I think they are very important points. So below is a summary of my other opinions.

The authors focus on lick–bout initiation and termination, and make the central claim that the lick cycle is controlled by the CPG in the brainstem, and that the cerebellum contributes to its initiation and termination.

This is a great improvement over the previous version. In particular, the detailed analysis shows that the cerebellar Crus I and II encode individual Licks, Lick rates, and information about lick–bout initiation and termination, which is a very interesting and exciting finding. The authors focus on lick–bout initiation and termination, and make the central claim that the lick cycle is controlled by CPG in the brainstem, and that the cerebellum contributes to its initiation and termination. But the experiments of optogenetics do not fully demonstrate the causality of this central claim. The story focusing only on the current central claim may be somewhat unreasonable.

Again, considering that their central claim is that the Lick cycle is controlled by the CPG in the brainstem, and that the cerebellum contributes to its initiation and termination, the optogenetic experiments performed to demonstrate causality do not clearly support this. First, regarding the initiation, considering that the neural activity in Figure 4 is a rising ramping activity, one would expect that the PC stimulation in Figure 7F would help the ramping and accelerate the initiation time. However, the result was rather the opposite. During stimulation, no initiation occurs, and in fact, it appears to strongly inhibit the initiation of licking (also in Figure 8B). Rather, initiation occurred as a rebound after stimulation. It needs to be properly explained why this is the opposite.

Next, regarding termination, considering that Figure 5 also shows ramp–up activity before the termination, PC stimulation at the peak of lick rate in Figure 7D would be expected to accelerate the termination time. Although this terminated the lick cycle as expected, licking started again after the stimulus offset. Then, the possibility remains that the stimulus only temporarily suppressed individual licking bouts, but did not terminate the lick cycle.

Figure 8 seems to have nothing to do with their central claim. I think this is a different story: The initiation time of the Lick cycle influences the subsequent motor plan (Licking peak time), as they stated (L557). This is interesting in itself, as it is related to Figure 2E, but I feel that the change in story here is abrupt, as the story has been following the central claim up to this point. I think some readers may find it hard to follow. I think it would be better if the story were more consistent.

L557 "a delay or discoordination in the transition to movement initiation, which is normally signaled by ramping PC activity, disrupts and/or delays the remaining motor plan, leading to a mistimed action."

As for the CF signal in Figure 6, it does not encode individual licks, which seems inconsistent with the Nature paper by Welsh et al. 1995. It would be good to have a discussion on why the different results were obtained.

Related to a concern of Reviewer 1, when calculating the firing rate, the firing rate for the period immediately before the event of interest is set to 0. Therefore, the 0 value is different in each plot, and the activity immediately before the event is missing. It is preferable to display the plots with the spontaneous firing rate set to 0 (at least in Supplementary Figures).

[Editors' note: further revisions were suggested prior to acceptance, as described below.]

Thank you very much for revising your article "The cerebellum encodes and influences the initiation and termination of discontinuous movements". We have discussed the revised manuscript with the original reviewers. All three agreed that the manuscript has improved significantly and contains important information which should merit those studying cerebellum and motor control.

However, during the consultation, it was brought to our attention that some statements regarding the optogenetic experiments are inappropriate. Purkinje cells' activity could be classified into two types:

1. PC activity for individual licking movements (as shown in Figure 3).

2. PC activity for the transition (initiation and termination) of licking cycles.

Therefore, the optogenetic stimulation of PCs should influence both individual licking movements and the transition of licking bouts. In the current manuscript, most statements focus on #2 without mentioning the influence of #1. It gives an impression of an over–interpretation or over–simplification and may lead to misunderstanding. Below are some examples:

– L38 (Abstract)

Optogenetic perturbation of PC activity disrupted the behavior in both initiating and terminating licking bouts, confirming a causative role in movement organization.

– L536 (Results)

Together, these results indicate that PC photostimulation can both initiate and terminate bouts of licking, depending on the timing of the activity perturbation relative to the licking context, indicating a role for PC activity in coordinating motor event transitions.

– L717 (Discussion)

Our optogenetic experiment provided causal evidence that Crus I and II PC activity influences movement performance.

– L741 (Conclusion of Discussion)

In summary, PC activity both represents and causes motor–event transitions, influencing the coordination of explicitly timed, volitional movements to improve the temporal consistency across repeat cycles of goal–directed action.

The above examples can be read as such that the causality between PC activation and behavior transition is clear (#2). However, this is not the case because activity #1 is also involved. Please take a moment to consider this point and clarify the statements to avoid confusion. Below are some recommendations from one of the reviewers.

1) Describe activity #1 clearly in the abstract and others. Activity #1 should be written with respect to previous works (e.g., Bryant et al.). Novel aspects of this study, such as analysis, etc., should be described more specifically.

2) The optogenetics experiment should not be discussed by solely focusing on the causality between activity #2 and behavior but should consider the causality between activities #1/#2 and behavior (rewrite L505).

3) Describe the conclusion of the optogenetics experiment, as "the experiment indicates that activities #1 and #2 contribute to the behavioral performance of Licking."

4) As a Discussion of the optogenetics experiment, it should be stated that the impairment may be mediated by changes in either Activity #2 (especially the experiment with photostimulation during Licking) or Activity #1 (especially experiments with photostimulation during the licking initiation) or both. The authors may discuss a possible approach to separate the two: it may be necessary to develop new paradigms such as pathway–specific photostimulation (individual stimulation of CPG–projecting PCs and non-CPG-projecting PCs).

When submitting your revised manuscript, please reinstate the previous Figure 4 —figure supplement 1. It is important to include this figure as a supplementary material because it is the only plot that shows the firing rates of the populations before subtracting the baselines.

We would also be grateful if you could give the title and abstract careful consideration. Please include in the title, a clear indication of the biological system under investigation. The abstract should not contain specialist abbreviations and acronyms where possible.

---

## [Author Response]

Essential revisions:1) The authors show that the simple spikes but not complex spikes ramp up before the initiation and termination of lick bouts. While this finding is important, the causality between the simple spikes and motor initiation/termination is unclear. In general, simple spikes fire more when the lick rate increases and confounds the interpretation of ramping activity (Reviewer 2). Furthermore, the trained licking behavior may not be well-timed enough to obtain a clear correlation between neuronal activity and behavior (Reviewer 1). These issues need to be clarified either by additional experiments or analysis.2) The results from the optogenetic experiments did not sufficiently support the main conclusion. Because the light stimuli by itself caused significant changes in licking, the shift in the lick–peak may have been induced indirectly by this abnormal licking behavior and not directly by the changes in simple spike activity (Reviewer 1-1, Reviewer 2-4). In addition, because optogenetic stimuli did not specifically change simple spikes, the involvement of complex spikes cannot be ruled out (Reviewer 3).

We have made extensive changes to the manuscript based on the reviewers’ recommendations. This includes an attempted to better characterize the relationship between Purkinje cell activity and behavior. We now demonstrate that Purkinje cell activity has a temporally nested structure and is modulated on the timescale both of individual lick cycles and at the initiation and termination of licking bouts. This is reported in a new main figure. We also addressed important points regarding expectation signals and Purkinje cell activity (new supplemental figures). Further analyses are used to address important concerns regarding optogenetic stimulation experiments and their effect on behavior (new supplemental figure). Last, we have also substantially altered the text to clarify our results and include important information regarding alternative possibilities to our conclusions.

Reviewer #1 (Recommendations for the authors):Specific concerns:1) It is difficult to interpret the optogenetic perturbation experiments in absence of corresponding neural data. While recording during such experiments certainly imposes an additional experimental burden, it is necessary to understand what essential features of PC activity dictate behavioral performance. In particular, it is striking that the same stimulation only produces a delay in peak licking when licks are initiated less than one second from the stimulus. Corresponding neural data revealing what differs between trials with short and long latency lick initiations would enable a much stronger interpretation of this result.

As the reviewer points out, we found that optogenetic PC activation just prior to lick bout initiation only influences the timing of the ensuing peak licking rate when the perturbation precedes this behavioral epoch by <1 sec. This implies a limited temporal correspondence between PC preparatory activity and the resulting behavioral adjustment. Notably, this result is congruent with the cerebellum’s role in organizing behavioral timing over short periods. For longer intervals (i.e., >1 sec), it is likely that neural activity has time to evolve/reset to its normal trajectory such that the peak licking rate was not delayed.

This may be attributable to the interceding activity of other brain regions (i.e., the basal ganglia or the cerebral cortex). We have edited the text to make this logic clearer. Making multi-site recordings and/or additional optogenetic perturbations to test this hypothesis is beyond the scope of the current project.

Related to this concern, Figure 7 demonstrates that optogenetic stimulation initiates licks on some (but not all) trials immediately following stimulation. The spike above baseline in lick initiation probability (Figure 7G) occurs < 1 s following optogenetic stimulation. This may suggest that the delay in peak lick rate only occurs when non–voluntary licks are initiated too close to the target time (i.e. on the trials where cessation of optogenetic stimulation drives immediate rebound licking)? The sub vs supra–second timing result could therefore relate to 1) the animals' voluntary initiation of licking too close to the optogenetic perturbation on a subset of trials, or 2) the animals' immediate involuntary response to cessation of optogenetic stimulation on a subset of trials. These two possibilities may suggest different interpretations for how the cerebellum contributes to the behavior.

We agree this is a concern. For this exact reason, we focused our analysis on the subset of trials *without* licking during the photostimulus period (note how the licking rate is pinned to zero). We apologize for not making this point clearer. That said, we were motivated to addresses the reviewer’s concern further, so we separately analyzed the subset of trials *with* licking during the photostimulus period. A response delay was also observed in these trials, like that for the remaining trials without licking during the photostimulus period. By ruling out the possibility that licking itself during the photostimulus period somehow disrupted subsequent licking around the time of water allocation, we conclude that the delay is attributable to the PC activity perturbation. This new analysis is included as a new supplemental figure and the text has been edited to clarify these points.

2) It is not clear that the observed responses relate to motor initiation and termination. For example, considering figure 4C vs 4E, the Sspk ramping at lick offset is quite different for omission trials as compared to rewarded trials. If these signals primarily reflect a timing response for stopping the motor program, they should not depend so heavily on reward context. Alternatively, the initiation signals could also be interpreted in the context of expectation. This is very challenging to disentangle, however. In this behavior, the omission trials do not serve well to do so, as it is not possible to align the data appropriately. Figure 2C shows that the animals have a poor temporal estimation of reward delivery on average, and maintain near–peak licking across an approximate 5 second window in absence of reward. Thus, because animals do not have a strong estimation of when the reward will occur, alignment to 'expected reward time' on omission trials is unlikely to capture any underlying responses. Related to this, given that Wagner et al. showed that granule cells also have reward omission responses in Crus I – it is important to reconcile this difference, even if it is simply because omission responses may not be identifiable given this task design.

While it is difficult to fully discount the possibility that some PC activity during the task may reflect a representation of expectation, evidence argues against that it fully accounts for the activity we observed around lick bout initiation and termination. First, the differences in ramping activity just prior to lick bout termination likely stem from differences in the licking behavior between the two contexts. Specifically, the licking rate is substantially lower on average for water omission trials than for water allocation trials. Therefore, the termination response is apt to be smaller. Second, if PC activity was mainly related to the context of expectation, then, on the next trial after water omission, PC activity would likely be disrupted by the prior expectation error. However, the average spiking activity of the Crus I PC ensemble in these trials (i.e., an *unexpected* 20 s interval) looks essentially identical to the pattern observed during trials of regular water allocation (i.e., the *expected* 10 s interval they were trained on). This result implies that PC activity likely encodes parameters related to licking performance rather than expectation. We have articulated these points in the manuscript and include an additional supplementary figure to directly address this concern.

Also related to this concern, the discussion argues that "our results reinforce the idea that the cerebellum influences well–timed, regularly performed actions that are generated solely by motivation...". This may not be accurate, and illustrates a major confound in interpreting signals during this behavior. Certainly, naïve animals are driven solely by motivation. They do not, however, exhibit any predictively timed licking behavior. In contrast, trained animals have harnessed the reward–related sensory stimulus to generate a new predictive behavior. This difference between naïve and trained animals shows that reward does double duty in this task – it is both a sensory stimulus that has intrinsic positive valence (and is thus sought solely by motivation in naïve animals), and a sensory cue that can guide temporal learning. Because naïve animals exhibit the former but not the latter, these two roles are separable. However, this dual role of the sensory input in this task also makes the interpretation of the neural responses in the learned condition challenging. In other words, because the cue and reward are one in the same in this task, it is extremely difficult to disambiguate sensory, expectation/prediction, and motor related signals. I am concerned that the task design may preclude causal relationships between the neural activity and behavior as a result.

We added to the discussion to highlight the reviewer’s points. We also removed the phrase “generated solely by motivation” in the text.

Additional questions:1) What is the depth of silicon probe recordings (what is the range of depths of the primary contacts for recorded PCs?). Does this overlap with imaging depths, or does the imaging data come from different regions than the electrophysiology? Do the Cspks recorded with electrophysiology exhibit the same behavior as those captured with calcium imaging?

Our electrophysiology and imaging approached sampled similar areas of Crus I and II because we performed the same, large craniotomy for both approaches (i.e., all animals were prepared the same way, irrespective of the recoding method, with the only difference being that we removed the cranial window prior to electrophysiology recording). We targeted a depth of <500 µm for silicon probe placement, which should correspond to the surface layers though it is difficult to truly ascertain because of deformation of the brain during penetration. Numerous prior reports have indicated that dendritic calcium events in Purkinje cells are an exacting proxy for complex spikes. Therefore, we elected to measure climbing-fiber-evoked responses optically. We have added additional information to the text to address these concerns.

2) What is the definition of a lick bout? Why is the average absolute lick rate (in s–1) pinned at 0 with no error for 2 seconds prior to lick bout initiation (e.g. Figure 3A)? It seems that bout initiations begin about 2.5 seconds prior to reward (Figure 3A bottom). Figures 1 and 2 show continuous licking on average for 5 seconds prior to reward. Doesn't this mean that there should be some non–zero lick rate prior to bout initiation? Alternatively, if these are licks defined by 2 seconds of preceding quiescence, shouldn't the initiations in the bottom of figure 2A bottom be further from the reward time?

To quantify lick-bout initiations, we identified well-isolated bouts so there was no ambiguity regarding when one bout stopped and the next started. Therefore, we set an arbitrary time in which the start of licking must be separated from other licks (i.e., absent of licks). We did not state this clearly, so we apologize for this omission. As the reviewer acknowledges, this period was >2 s long for the data presented in the figure. Hence, licking was pinned to 0, without error, for 2 s before initiation in lickaligned bouts (notably, this definition meant that there were uncategorized licks which were not included in the lick-aligned analysis). Also, the reviewer should note that the in the top of figure panel A, licking is aligned to the first lick. Whereas in the bottom, it is aligned to the time of expected water delivery. To address the reviewer’s concern, we have defined what a lick bout is in the text. We also state that we only included well separated bouts, based on the criterion described above, in the analysis.

3) It looks like there is a Cspk response to the last lick in figure 6I – is this significant? Is this response different between rewarded and reward–omission trials?

A significance test is now reported in the manuscript. There is no difference between rewarded and reward-omission trials.

4) How are lick initiations defined surrounding optogenetic stimulation? Optogenetic stimulation suppresses licking, and results in rebound responses afterwards. When these occur in the pre–water period, they are defined as initiations. Why aren't the post water licks that follow the quiescence period of optogenetic stimulation considered initiations? Would not the same analysis show in 7G (related to 7F) yield a similar spike in initiation probability if performed on the data from 7D?

For the dataset in the figure panel, we used the same criteria to define lick bout initiations surrounding optogenetic stimulation as for the interleaved control trials (the blue and black lines, respectively). Again, we arbitrarily used a 2 s period of non-licking to define well-separated lick bouts. As this was the period of earliest lick-bout initiations, there were plenty of bouts for inclusion. In contrast, the rebound in licking after the optogenetically induced period of suppression in figure panel D would not qualify as well separate initiations based on our criterion because this period of suppress licking was short lived, lasting several hundred milliseconds. As noted in the text, the licking pattern during the optogenetic perturbation was highly disturbed: rhythmicity was disrupted with the licks appearing erratic and mostly ceased altogether. Shortly after, licking resumed but also appeared irregular regarding both variance and rate. Therefore, it was difficult to categorize what type of licking this was (i.e., whether it was an initiation or a continuation of licking). Due to this ambiguity, we did not perform the same analysis for this dataset as we did for figure panel F. We have added additional descriptive information in the text regarding our criterion for lick bouts to address this concern.

5) It's a bit odd to argue that imaging was necessary because it is "challenging to reliably distinguish complex spike waveforms" (line 337), while at the same time making the case that cell sorting was validated based on the property that PC units were 'unambiguously identified by accompanying complex spikes in the recordings" (line 774).

By stating that it is "…challenging to reliably distinguish complex spike waveforms…", we did not mean to imply that we could not detect any complex spikes. Rather, we were trying to relay our lack of confidence in identifying all of them. For unambiguous PC classification, we only needed to identify SOME complex spikes in our unit recordings. However, to analyze how complex spikes represent behavioral attributes, we believed we needed to identify most of them. Furthermore, because imaging allowed us to measure calcium transients in many Purkinje cells simultaneously, we were able to increase the sample size significantly beyond what would have been feasible with electrophysiology. To address the reviewer’s concern, we have edited the text to make this point.

Reviewer #2 (Recommendations for the authors):I recommend the following analysis, presentation, and discussion to improve the paper.For the first major comment, the authors need to evaluate the encoding of each bout of licking by cross–correlation or event triggered averaging, and separate it from the encoding of initiation and termination (e.g. using linear model).

Cross-correlation analysis of each lick bout requires some variability in the statistics of the behavior. Unfortunately, the intervals between licks, when examined across session-trials from individual animals, are extremely regular, precluding single-bout analysis of lick rate and PC activity. For this reason, we had to examine all licking across all animals to see any correlation. We considered using linear models as the reviewer suggested to address how the broader behavior (i.e., each bout) is encoded by PC activity. However, such models require distinct variables to be present and absent in different trials. Because every bout of licking includes a start and stop to the action, we could not parse trials into those that include lick-bout initiation but not termination, and vice-versa. Linear models, therefore, are not appropriate.

To address the reviewer’s concern, we performed a new analysis focused on the representation of individual licks in PC firing. Our new results (Figure 3), show remarkable heterogeneity in lick-phase encoding in the PC ensemble. Important to the reviewer’s concern, the entrainment strength of PC activity to individual licks, was not predictive of their overall firing pattern when examined on a cell-bycell basis. This is consistent with a view of a rich representation of nested behavioral variables at different timescales for periodically performed, discontinuous movements.

For the second comment, they should further analyze the negatively modulating Purkinje cells, shown in Figure 5C. They also need to display the histology of their recording sites (Schematic drawing in Figure 2A is not enough). Also, they should discuss about the potential bias in recording of Purkinje cells.

In our dataset, there are not enough negatively modulating PCs for further analysis. We were hesitant to perform additional experiments in a concerted attempt to only identify negatively modulating PCs because this makes an a priori assumption of what type of PC activity is important for task performance. Histological confirmation of our recording sites was unnecessary because we could clearly target our probes to either Crus I or Crus II under visual guidance. As mentioned above, our craniotomy site was large (atypical for in vivo electrophysiology recording), exposing the surfaces of both Crus I and Crus II which are clearly distinguishable from one another based on the sulcus dividing them, allowing targeted recordings to either lobule under investigator choice. These are the same areas that we have recording from for the past 5 years with little ambiguity (see Gaffield et al. 2016, 2017, 2018, and 2019). We have now specified this logic in the text. Last, as mentioned above, we have extensively edited our text to discuss how our results fit in with a scheme of antagonistic cerebellar modules.

For the third comment, I recommend the spike rete–based display for the data presentations, which should include the spontaneous spike rates.

As mentioned in our response above, we chose to show z-scored spike rates as this is the current standard in the neurophysiology field. That said, we changed how we displayed the activity of individual cells as spike rates.

For the fourth and fifth comments, related to the first comment, if the Purkinje cells contribute to controlling each of licking bout, the observation in the optogenetic experiments could be explained. The authors need to consider the possibility and improve the interpretation and conclusion.

We again apologize for not emphasizing how PCs encode additional behavioral variables related to an online representation of the licking bout itself. We have edited the text to make this point clearer and have elaborated on this point to improve the interpretation and conclusion of our optogenetic results with this in mind.

Reviewer #3 (Recommendations for the authors):– Figure 3 E shows the difference in simple spike onset between pre and post licking. It seems both Crus I and II were pooled together, but this analysis should be conducted separately in different lobules.

The reviewer is correct, PC activity from both Crus I and Crus II were originally pooled together because the effect of early ramping onset times for anticipatory licking was similar. However, in response to the reviewer’s suggestion, we now present the analysis separately for each lobule.

– Figure 3 F shows representative PC activity relative to the first lick. Is this trace from pre–water lick or post–water lick? Representative traces from both pre–water and post–water lick should be presented to demonstrate the difference in the ramp onset.

Figure 3F showed activity from three representative PCs aligned to pre-water licking bouts. We now include PC activity from post-water licking bouts; this is presented in a supplemental figure. The timing difference in PC activity relative to the types of licking are starkly evident in these examples.

– Figure 4C shows spike rate changes before the termination of the lick. As the firing rate in Figure 3C–D is plotted in (s–1), this data should be also presented using the same unit for comparison.

Figures 3 and 4 have been edited for continuity as recommended by the reviewer. We chose to show z-scored spike rates as this is the standard metric for reporting activity profiles across cells.

– Figure 5D, please revise color coding for easier interpretation.

We changed the color scheme of this figure panel for easier interpretation.

– Figure 5D shows a significant number of cells in Crus II ramped up for both pre–water and post-water licking. It would be informative to compare the ramp onset between pre-water and post–water licking for these cells. The result will show whether the same cells responded differently to the self–initiated and externally triggered licking.

We identified 8 PCs that showed ramping activity for both pre- and post-water licking. Some cells had clear timing differences between the two different licking contexts, ramping earlier for anticipatory licks compared to reactive consummatory licks (see PC1 and PC2 in the new supplemental figure). Across these 8 cells, the onset time of activity ramping relative to pre- and post-water licking was -0.300 ± 0.114 ms and -0.1125 ± 0.242 ms, respectively. Although there was a clear trend in the data, the difference was insignificant (p = 0.07; Student’s t-test). This implies that PCs tuned to either pre- or postwater licking also contributed to the overall timing difference observed in the entire PC ensemble.

– In Figures 7 and 8, optogenetic stimulation of PCs was conducted with ChR2. While this experiment was performed to test the involvement of simple spikes, it could be mediated by alteration in complex spikes. Were complex spikes affected by the stimuli? Representative traces from raw data for Figure 7A should be presented to address this point.

As mentioned in the text, we have recently shown that direct optogenetic activation of PCs can elicit responses resembling complex spikes. However, this effect is dependent on the intensity of the optogenetic stimulus (Bonnan et al., 2021). Although we used a consistent light power for the photostimulus across animals, we cannot control for the inevitable light-power variability within the stimulation site (e.g., the edges of the optical fiber tip or with depth), even if we recorded PC activity simultaneously with the optogenetic stimulus. Therefore, we cannot fully discount the possibility that complex-spike-like bursts were elicited in some PCs, in addition to changes in simple spiking when ChR2expressing PCs were photostimulated. In the end, we want to be conservative regarding our interpretation of the effect of ChR2-induced stimulation of PCs. We have edited the text to make this rationale clearer.

– Figures 8B and 8F show the delayed peak in lick rate with PC activation. This finding should be supported with additional statistical analysis.

We now include statistical tests for the delay in lick rate with PC photostimulation. Note that the timing differences shown in Figure 8E and 8F are summarized in Figure 8G, with accompanying statistical tests (asterisks denote a significant timing difference relative to control).

[Editors' note: further revisions were suggested prior to acceptance, as described below.]

Reviewer 1:I have several comments below directly related to our initial reviews. I think the paper is interesting, and that they could revise one more time with no experiments to address these issues.The main concerns about the initial submission focused largely on the question of whether the authors' data and analysis support the central claim that the measured neural activity patterns are causally related to motor initiation and termination. In response to these concerns, the authors have provided several important clarifications and some new analyses. While I remain positive about the potential impact of the findings in this study, and would support this publication for eLife, there are some important remaining issues directly related to those central concerns.1. Now that the authors have clarified the analysis related to figure 4, I am concerned about its main conclusions. Specifically, the authors state that "the average simple spiking rate began to ramp earlier for exploratory licking trials, when the movements were initiated prior to water allocation, compared with that for trials in which consummatory licking commenced immediately after water became available" (lines 205–208).This is a key point meant to distinguish neural responses on planned vs unplanned movements, but I am not convinced by the underlying analysis. The authors have subtracted z–scored neural data for pre and post water licks, and used the difference to suggest that pre water licks have an associated neural responses that "starts earlier". However, this analysis conflates response amplitude and timing, as a subtraction only reveals when responses diverge (the negative latency in the difference trace cannot be taken to mean that the larger response began earlier). To specifically address timing (and not amplitude), the responses must first be scaled, and then subtracted. Because a smaller response takes longer to cross the same amplitude threshold, a subtraction will show a difference right away, even if the responses start at the same time. In addition, the quantification as performed in Figure 4F will necessarily indicate shorter latencies for larger responses.It seems very likely that a scaled subtraction will show no latency difference for the data in figure 4C and D, and this would contradict the conclusion of a timing difference in neural responses for planned and unplanned movement.

Provided the reviewer’s concern, we have removed z-scoring from the paper. The timing difference between the two licking contexts is apparent in the plots of population activity, now represented as the change in firing rate. The differences are rigorously quantified in the positively modulating PCs (i.e., those that account for the ramping in the population response) in Figure 4G and H (the subtracted z-score panels were always for illustration purposes). Regarding Figure 4F (now 4H), this analysis is based on an assessment of the simple spiking pattern of individual PCs (deviation from baseline that was independent of amplitude), so the reviewer’s concern is not applicable.

The data in Figure 4 do, however, show a much smaller response on unplanned movements, which is interesting. However, it is unclear whether or not the licking is different in this case? This needs to be shown for the associated neural traces in figure 4. The overall data suggest that lick bouts are relatively homogenous at their initiation. Thus, these data may indicate that the amplitude of the neural response does not reflect lick rate, and the timing of the response does not relate to prepared or unprepared movement. As reviewer 2 notes, PC activity also does not seem to represent peak lick rate timing.

As we showed in Figure 6, water-reactive (post-water) licking bouts have a higher peak rate of licking compared to exploratory (pre-water) licking bouts. This result is now presented in Figure 4. We did not find a significant change in the peak simple spiking response around the time of peak licking between the different types of licking. This likely owes to the relatively small difference in peak lick rate between these two contexts (about 1 s^-1^). As shown in Figure 2—figure supplement 2, differences in PC simple spike firing are only apparent with large differences in licking rate (e.g., between 3 and 6 s^-1^). These comparisons and discussion points have been added to the manuscript.

Together with the above, I therefore remain unclear on the authors model linking neural activity and behavior, and whether the recorded activity relates to motor preparation / planning / execution of some kind.Other important issues:2. In reading the other reviewers concerns about z–scoring and the authors responses, I realized that the z–scoring is performed to different baselines for different analyses, and not to the mean spike rate across the trial or a common reference period across analyses (the specifics of the z–scoring are not in the methods). This can be seen in difference between figures 4C and 5C, for example. I am concerned about this practice for a couple of reasons:2.1) it means that the amplitude of neural signals cannot be compared across different figures and conditions. This makes it challenging to interpret the relationship between firing changes and behavior.

As mentioned above, to facilitate comparisons between figures, we removed z-scoring. Instead, we use changes in PC simple spike firing rate. Of course, this comes at the cost of not having an inherent representation of the significance of the change, which is the advantage of z-scored responses.

2.2) to disambiguate other explanations for the neural data in this paper, it would be helpful to further leverage conditions where behavior differs. For example, in Figure 4C, there are trials where no licks occur until after presentation of water. This affords the opportunity to ask whether or not there was any increase in spike rate before water allocation in absence of licking (and thus test whether changes in spike rate might reflect something other than motor initiation). However, the authors have z–scored to the mean immediately preceding post–water licks, which would obscure any such changes (Figure 4 supp. 2 may show such an increase pre–water?). While I am sympathetic to the authors' arguments that z–scoring is commonplace for neural recording data, the implementation here is not ideal for evaluating the relationship between spiking and behavior.

For our new analysis, we calculated baseline from the activity level measured across all non-licking periods for each cell during recording (i.e., we did not baseline immediately before the onset of licking). We did not observe an increase in spiking activity prior to water allocation in post-water licking. For plots in Figure 5, we also show activity baselined to the activity during the preceding licking. We believe this facilitates comparisons across figures. To further address the reviewer’s point regarding nonmotor signals, we also analyzed the change in PC firing in reward-absent trials without licking. There was no response; this new data is included as a new supplemental figure. We also now cite the findings of Sendhilnathan et al. 2020 who identified widespread movement-related activity, but not reward or reward expectation activity (at least in the absence of learning), in Crus I and II PCs of monkeys performing an overtrained motor task. We think our results are congruent with this observation.

3. In the first round of reviews, the question of how to appropriately disambiguate expectation based on omission trials was raised, given the animals imprecise expectation of the time of water delivery. This concern necessitates a more convincing analysis in order to support the authors statements in the discussion regarding expectation signals. For example, with complex spiking on omission trials, alignment to the first lick after the time of water expectation would provide a more appropriate timepoint to indicate the animals' expectation. Even better would be to look at the moment when lick bouts start to decrease / are terminated on omission trials, as this is the timepoint when the animals' behavior indicates recognition that the expected reward is not present (and this time has previously been shown to reveal such expectation signals, at least in some conditions). There may indeed be no evidence of expectation signals in this behavior, but in absence of such analysis to evaluate the question appropriately, it seems premature to make such conclusions.

We previously attempted this analysis. However, for climbing-fiber-induced PC activity, there were too few omission trials to generate a clear result.

4. This point is only a suggestion, but I think others may have the same confusion regarding figure 8. The difference between 5 and 10 second trials is not that licking ramps up more slowly on 10 second trials – rather, the mean lick plots shown here speak to the probability of lick bout initiation across trials. On single trials, licking just goes from nothing to the patterned bout rate. On average, however, this manifests as a ramp due to variability in the onset of lick bout times, and the increase in probability of initiation as the trial progresses. The essential feature of 10 second trials is that lick bouts start on average later than they do for 5 second trials. This necessarily means that lick bouts are more likely to be outside of the 1.25 second window from optogenetic stimulation defined by the analysis in 8F. However, if the same analysis from figure 8E and F were performed for 10 second trials (<1 vs > 1.25 second initiations from stimulation), it should yield the same result. It is likely that the average licking for 10 second trials following stimulation only looks different because it is weighted so much more heavily toward bouts initiated >1.25 seconds from stimulation.This analysis would greatly enhance the authors point that these effects are all about the duration from optogenetic stimulation when the animal tries to lick, and not about the absolute duration of the trial (as many will likely assume based on the figure and its description). Perhaps this seems obvious, but such clarification could provide strong support for the authors arguments related to the final figure.

At the urging of Review 2, we removed Figure 8. The reviewer’s point is now moot.

Reviewer 2:The paper contains very interesting topics and findings and has a potential impact worthy of eLife, but I think it is still too immature to be published with minor revisions. I think the opinion of revising it again without experiments is very valid.I also agree with the concerns of Reviewer #1. In particular, I have exactly the same concerns about the main concern and #2 issue, and I think they are very important points. So below is a summary of my other opinions.The authors focus on lick–bout initiation and termination, and make the central claim that the lick cycle is controlled by the CPG in the brainstem, and that the cerebellum contributes to its initiation and termination.This is a great improvement over the previous version. In particular, the detailed analysis shows that the cerebellar Crus I and II encode individual Licks, Lick rates, and information about lick–bout initiation and termination, which is a very interesting and exciting finding. The authors focus on lick–bout initiation and termination, and make the central claim that the lick cycle is controlled by CPG in the brainstem, and that the cerebellum contributes to its initiation and termination. But the experiments of optogenetics do not fully demonstrate the causality of this central claim. The story focusing only on the current central claim may be somewhat unreasonable.Again, considering that their central claim is that the Lick cycle is controlled by the CPG in the brainstem, and that the cerebellum contributes to its initiation and termination, the optogenetic experiments performed to demonstrate causality do not clearly support this. First, regarding the initiation, considering that the neural activity in Figure 4 is a rising ramping activity, one would expect that the PC stimulation in Figure 7F would help the ramping and accelerate the initiation time. However, the result was rather the opposite. During stimulation, no initiation occurs, and in fact, it appears to strongly inhibit the initiation of licking (also in Figure 8B). Rather, initiation occurred as a rebound after stimulation. It needs to be properly explained why this is the opposite.Next, regarding termination, considering that Figure 5 also shows ramp–up activity before the termination, PC stimulation at the peak of lick rate in Figure 7D would be expected to accelerate the termination time. Although this terminated the lick cycle as expected, licking started again after the stimulus offset. Then, the possibility remains that the stimulus only temporarily suppressed individual licking bouts, but did not terminate the lick cycle.

We did not mean to imply that the lick cycle is under the sole control of a brainstem CPG. Likely, multiple brain regions cooperate to determine the performance of this motor behavior. In fact, we show that the lick cycle is encoded by cerebellar activity and its perturbation can affect rhythmicity. We focused on lick bout initiation and termination because these parameters are poorly understood. As explained in our prior rebuttal, the licking behavior that we studied was quite regular across trial conditions. Due to the low amount of variability in the licking response, it is difficult to draw conclusions between changes in PC firing and licking rate on a trial-by-trial basis which would provide insight to the reviewer’s point. Certainly, this will be the goal in a subsequent study.

In about half the trials, the mice did not resume licking after stopping in response to the optogenetically induced perturbation of PC activity. This result is now reported in the text. We believe this is in line with a robust “stop” to action rather than a temporary pause of the behavior. In trials where the mice resumed licking, it is impossible for us to disambiguate whether this is a stop-restart action rather than a true pause in the behavior as the mice are likely to continue to be motivated to consume water rewards after the optogenetically induced cessation of licking and therefore reinitiate licking shortly after they had abruptly stopped.

Figure 8 seems to have nothing to do with their central claim. I think this is a different story: The initiation time of the Lick cycle influences the subsequent motor plan (Licking peak time), as they stated (L557). This is interesting in itself, as it is related to Figure 2E, but I feel that the change in story here is abrupt, as the story has been following the central claim up to this point. I think some readers may find it hard to follow. I think it would be better if the story were more consistent.L557 "a delay or discoordination in the transition to movement initiation, which is normally signaled by ramping PC activity, disrupts and/or delays the remaining motor plan, leading to a mistimed action."

We eliminated this figure based on the reviewer’s urging.

As for the CF signal in Figure 6, it does not encode individual licks, which seems inconsistent with the Nature paper by Welsh et al. 1995. It would be good to have a discussion on why the different results were obtained.

We did not make this claim. In fact, we previously examined CF-induced activity in Crus II PCs aligned to individual licks in water-consummation bouts and found a result like Welsh et al. 1995 (see Gaffield et al., 2016). In our current manuscript, we focused on CF activity aligned to the initiation and termination of licking which Welsh et al. did not do. We now reference this prior work in the discussion.

Related to a concern of Reviewer 1, when calculating the firing rate, the firing rate for the period immediately before the event of interest is set to 0. Therefore, the 0 value is different in each plot, and the activity immediately before the event is missing. It is preferable to display the plots with the spontaneous firing rate set to 0 (at least in Supplementary Figures).

For our plots showing the change in PC firing rate, baseline is determined from the activity level for all non-licking periods during the recording. We believe this meets the reviewer’s criteria of displaying plots with “spontaneous” firing set to 0, though that definition of “spontaneous” is subjective in a continuously behaving animal.

[Editors' note: further revisions were suggested prior to acceptance, as described below.]

However, during the consultation, it was brought to our attention that some statements regarding the optogenetic experiments are inappropriate. Purkinje cells' activity could be classified into two types:1. PC activity for individual licking movements (as shown in Figure 3).2. PC activity for the transition (initiation and termination) of licking cycles.Therefore, the optogenetic stimulation of PCs should influence both individual licking movements and the transition of licking bouts. In the current manuscript, most statements focus on #2 without mentioning the influence of #1. It gives an impression of an over–interpretation or over–simplification and may lead to misunderstanding. Below are some examples:

We thank the reviewers for their positive assessment of our revised manuscript and appreciate the importance of this last concern. We have addressed this point in our revised manuscript, as detailed below.

– L38 (Abstract)Optogenetic perturbation of PC activity disrupted the behavior in both initiating and terminating licking bouts, confirming a causative role in movement organization.– L536 (Results)Together, these results indicate that PC photostimulation can both initiate and terminate bouts of licking, depending on the timing of the activity perturbation relative to the licking context, indicating a role for PC activity in coordinating motor event transitions.

We edited both sentences to indicate that we also examined the effect of perturbing PC activity on cycles of individual licking movements.

– L717 (Discussion)Our optogenetic experiment provided causal evidence that Crus I and II PC activity influences movement performance.

This is a generic sentence that is inclusive of all our findings. The effect of PC activity perturbation on individual licking movements and motor transitions is separately elaborated on in the two sentences that follow it.

– L741 (Conclusion of Discussion)In summary, PC activity both represents and causes motor–event transitions, influencing the coordination of explicitly timed, volitional movements to improve the temporal consistency across repeat cycles of goal–directed action.

This concluding sentence been modified to be more inclusive of all our findings.

The above examples can be read as such that the causality between PC activation and behavior transition is clear (#2). However, this is not the case because activity #1 is also involved. Please take a moment to consider this point and clarify the statements to avoid confusion. Below are some recommendations from one of the reviewers.1) Describe activity #1 clearly in the abstract and others. Activity #1 should be written with respect to previous works (e.g., Bryant et al.). Novel aspects of this study, such as analysis, etc., should be described more specifically.

We edited the manuscript to describe PC activity related to individual licks in the Abstract and more thoroughly in the Discussion. We have previously specified our use of multiple modes of in vivo activity measurements and use of optogenetics, both in combination with analysis of a periodically performed discontinuous motor behavior, which makes our study unique. Regarding Bryant et al., we have repeatedly cited this manuscript by the Heck Lab throughout our manuscript to call attention to the importance of this work.

2) The optogenetics experiment should not be discussed by solely focusing on the causality between activity #2 and behavior but should consider the causality between activities #1/#2 and behavior (rewrite L505).

We have rewritten this sentence to be inclusive of both types of PC activity and how their perturbation could explain the optogenetic result.

3) Describe the conclusion of the optogenetics experiment, as "the experiment indicates that activities #1 and #2 contribute to the behavioral performance of Licking."

We added the statement that optogenetic perturbation of PC activity also effects individual cycles of licking.

4) As a Discussion of the optogenetics experiment, it should be stated that the impairment may be mediated by changes in either Activity #2 (especially the experiment with photostimulation during Licking) or Activity #1 (especially experiments with photostimulation during the licking initiation) or both. The authors may discuss a possible approach to separate the two: it may be necessary to develop new paradigms such as pathway–specific photostimulation (individual stimulation of CPG–projecting PCs and non-CPG-projecting PCs).

As requested, we have discussed this possibility. We also added that, in the future, targeting specific PC pathways (if they exist) may allow disambiguation of these effects.

When submitting your revised manuscript, please reinstate the previous Figure 4 —figure supplement 1. It is important to include this figure as a supplementary material because it is the only plot that shows the firing rates of the populations before subtracting the baselines.

We reinstated Figure 4—figure supplement 1 (now Figure 4—figure supplement 2).

We would also be grateful if you could give the title and abstract careful consideration. Please include in the title, a clear indication of the biological system under investigation. The abstract should not contain specialist abbreviations and acronyms where possible.

We edited the title to indicate that the experiments were performed in mice. We also updated the title to reflect the reviewers’ input outlined above.